# Semantic Segmentation of Agricultural Images Based on Style Transfer Using Conditional and Unconditional Generative Adversarial Networks

**Hirokazu Madokoro** [1,*], **Kota Takahashi** [2], **Satoshi Yamamoto** [3], **Stephanie Nix** [1], **Shun Chiyonobu** [4], **Kazuki Saruta** [2], **Takashi K. Saito** [2], **Yo Nishimura** [5] **and Kazuhito Sato** [2]

1   Faculty of Software and Information Science, Iwate Prefectural University, Takizawa 020-0693, Japan; nix_s@iwate-pu.ac.jp
2   Faculty of Systems Science and Technology, Akita Prefectural University, Yurihonjo 015-0055, Japan; m22a013@akita-pu.ac.jp (K.T.); saruta@akita-pu.ac.jp (K.S.); saito@akita-pu.ac.jp (T.K.S.); ksato@akita-pu.ac.jp (K.S.)
3   Faculty of Bioresource Sciences, Akita Prefectural University, Akita 010-0195, Japan; syamamot@akita-pu.ac.jp
4   Graduate School of International Resource Sciences, Akita University, Akita 010-8502, Japan; chiyo@gipc.akita-u.ac.jp
5   Agri-Innovation Education and Research Center, Akita Prefectural University, Ogata 010-0444, Japan; ynishimu@akita-pu.ac.jp
*   Correspondence: hirokazu_m@iwate-pu.ac.jp; Tel.: +81-019-694-2500

**Abstract:** Classification, segmentation, and recognition techniques based on deep-learning algorithms are used for smart farming. It is an important and challenging task to reduce the time, burden, and cost of annotation procedures for collected datasets from fields and crops that are changing in a wide variety of ways according to growing, weather patterns, and seasons. This study was conducted to generate crop image datasets for semantic segmentation based on an image style transfer using generative adversarial networks (GANs). To assess data-augmented performance and calculation burdens, our proposed framework comprises contrastive unpaired translation (CUT) for a conditional GAN, pix2pixHD for an unconditional GAN, and DeepLabV3+ for semantic segmentation. Using these networks, the proposed framework provides not only image generation for data augmentation, but also automatic labeling based on distinctive feature learning among domains. The Fréchet inception distance (FID) and mean intersection over union (mIoU) were used, respectively, as evaluation metrics for GANs and semantic segmentation. We used a public benchmark dataset and two original benchmark datasets to evaluate our framework of four image-augmentation types compared with the baseline without using GANs. The experimentally obtained results showed the efficacy of using augmented images, which we evaluated using FID and mIoU. The mIoU scores for the public benchmark dataset improved by 0.03 for the training subset, while remaining similar on the test subset. For the first original benchmark dataset, the mIoU scores improved by 0.01 for the test subset, while they dropped by 0.03 for the training subset. Finally, the mIoU scores for the second original benchmark dataset improved by 0.18 for the training subset and 0.03 for the test subset.

**Keywords:** annotation; semantic segmentation; smart farming; GAN; FID; IoU; CUT; pix2pixHD; DeepLabV3+

## 1. Introduction

Smart farming provides labor conservation, automation, and improved efficiency of farming operations, thereby leading to advancement of value-added agricultural activities [1]. Some representative case studies of successful smart farming are automatic operations of farm machinery in terms of tractors and combine harvesters using multiple sensors and global navigation satellite systems (GNSS), automatic water control and management systems based on internet of things (IoT) sensors and devices [2], crop monitoring,

and pesticide and fertilizer spreaders using drones [3]. Although smart farming has already progressed to the implementation, demonstration, and dissemination phases, the major tasks are limited to automated repetition of simplified operations and locomotion based on operational procedures and routes that must be set in advance.

Practical applications of deep learning (DL) [4] technologies for smart farming are progressing rapidly [5]. Compared with general DL applications in terms of automated driving [6], biometric identification [7], and medical image processing [8], technical challenges remain, especially for classification, recognition, and decisions for agricultural applications [9–11]. Moreover, the collection of high-quality and large agricultural datasets [12] entails great burdens because of the wide variations in crop growing periods, weather patterns, air temperature, seasonal changes, and various external factors [13]. Although extensive amounts of elaborately annotated data improve DL performance and avoid overfitting [14], the burdens and time of annotation [15], especially for pixel-wise segmentation labels, make acquiring and using them labor-intensive and expensive. Therefore, numerous research tasks remain for DL applications in smart farming [16].

This study specifically examines semantic segmentation, which is a major task in computer vision studies, to explore DL-based solutions as challenges for smart farming [17]. Particularly, this study examines semantic segmentation of crop images [18] related to disease detection, weeding, and cultivation management. Image segmentation can be of three types: semantic segmentation, instance segmentation, and panoptic segmentation [19]. Segmentation targets comprise things as countable objects and stuff as matter, material, and textures. Semantic segmentation can handle both targets. Things within the same object category are assigned the same label. Instance segmentation handles only things. Things in each object are assigned with unique labels for the respective instances. Panoptic segmentation includes both semantic and instance segmentation mechanisms.

Champ et al. [20] proposed an instant segmentation method for crop and weed plants using a region-based convolutional neural network (R-CNN) [21]. Instance segmentation provides flexible detection of each crop and weed for sparse vegetation. Their research project specifically examined sparse vegetation images because they were intended to be mounted on a precision agricultural robot. By contrast, instance segmentation will be challenging if the vegetation distribution is dense. This paper presents a semantic segmentation framework to exploit agricultural images with dense vegetation.

Numerous benchmark image datasets are available for semantic segmentation in agriculture [22]. These datasets were annotated using ground truth (GT) labels through an enormous effort. Annotations for agricultural datasets are conducted for the respective purposes and tasks. Special training is also necessary for annotators in advance, depending on objects such as crop damage and leaf types. To reduce burdens for annotating and producing synthesized data [23], this study is intended to generate crop datasets based on image style transfers [24] using generative adversarial networks (GANs) [25]. Results obtained from experimentation using the proposed framework demonstrate that the quantity and quality expansion of images and labels using GANs improve accuracy and convergence compared to existing semantic segmentation models using source images and labels.

This paper is structured as follows. Section 2 briefly reviews state-of-the-art data augmentation methods based on image style transfer and GANs. Subsequently, Sections 3 and 4 present our framework for semantic segmentation combined with data augmentation using two GAN models with different mechanisms and their setup properties, including benchmark datasets of three types and evaluation metrics. The preliminary and evaluation experiment results are presented in Sections 5 and 6. Finally, Section 7 presents the conclusions and explains some intentions for future work.

## 2. Related Studies

Existing data augmentation studies are divisible into two approaches. The first approach is based on model-based algorithms. Rozantsev et al. [26] proposed an algorithm

that synthesizes virtual 3D object images superimposed on specific background images. Rematas et al. [27] proposed an image generation method from unknown viewpoints using structural features extracted from 3D objects. They applied their method to data augmentation for creating super-high-resolution images combined with 2D to 3D transformations. Pishchulin et al. [28] proposed a data augmentation method for pedestrian detection and pose estimation. They specifically controlled the poses and appearances of the existing data. However, these data augmentation methods were applied to unsupervised classification tasks because the augmentation targets were merely source data without labels. As one example of research into automatic label generation, Vázquez et al. [29] detected pedestrians using labels generated from outdoor scene images that were created using a simulator. Their proposed domain-adapted pedestrian classifier was trained by combining a small number of pedestrian samples from a real-world target domain with numerous ones from a virtual-world source domain. Ros et al. [30] proposed a method to synthesize images and labels in a virtual urban environment. They demonstrated application experiments for semantic segmentation using synthesized multi-camera images from a virtual car after annotation, including different labels for each season.

The second approach is based on learning algorithms. This approach includes numerous methods based on conventional machine-learning (CMT) algorithms and DL algorithms using state-of-the-art backbones such as ConvNets [31–39] and Transformers [40–44]. Recently, numerous studies of GANs have been conducted for data generation and augmentation [45–49]. Fundamentally, GANs comprise a generator and a discriminator. A generator supplies virtual images generated from noise signals to a discriminator. The learning objective for a discriminator is the classification of real images and generated pseudo-images. The learning objective for a generator is the opposite of that of a discriminator. This relation is consistent with the mini-max principle [50]. A discriminator learns from both images while avoiding false segmentation of augmented images provided by a generator. The respective networks modify weights simultaneously and in parallel to opposite objectives as adversarial learning.

Radford et al. [51] proposed a deep convolutional GAN (DCGAN) for unsupervised representation learning. They introduced convolutional layers as an architectural topology inside of the generator. Based on the energy-based GAN (EBGAN) [52], Berthelot et al. [53] proposed a boundary equilibrium GAN (BEGAN) that uses a convolutional auto-encoder (CAE) [54] for a discriminator. Although DCGAN contained no mechanism to judge the convergence of learning for a generator and a discriminator, BEGAN solved this difficulty after introducing loss functions to the respective networks. Karras et al. [55] proposed a progressive growing GAN (PGGAN) that adds new layers according to learning progress. They demonstrated not only the creation of high-resolution images, but also a reduction in computational cost for training because their proposed method gradually learned the overall image structure.

As an alternative generator architecture based on the arbitrary style transfer mechanism [56], Karras et al. [57] proposed StyleGAN for learning unsupervised separation of high-level attributes and stochastic variation in the augmented images. They introduced not only a learning mechanism to increase resolutions resembling PGGAN progressively, but also local vectors to add probabilistic variation and global vectors to decide the object orientations and respective parts in the generative process. StyleGAN has been used as a baseline in numerous studies because of its high accuracy and suitability for various applications of image generation. Moreover, the process of image generation based on GANs includes practical applications [58]. Wang et al. [59] proposed an enhanced super-resolution GAN (ESRGAN) for applications that transform low-resolution images to high-resolution images. Zhang et al. [60] proposed a new model called StackGAN to convert text to images. These studies demonstrated the wide range of GAN applications, with research and development becoming more active in recent years.

GAN frameworks are also applied to image style transfer [61]. Zhao et al. [62] proposed a crop detection method using StyleGAN [57] combined with deep visual transfer

learning. We consider that data augmentation with image style transfer based on GANs contributes to the improvement of segmentation accuracy in agricultural applications. GANs can be of two models: conditional GANs and unconditional GANs [63]. This difference derives from the correspondence between source images and labels. On the one hand, unconditional GANs learn generative models from source images. On the other hand, conditional GANs learn generative models from source images and labels similarly to supervised learning. As a representative conditional GAN, Isola et al. [64] proposed pix2pix, which uses two conditional GANs to enable a conversion task between images. Wang et al. [65] proposed an improved model pix2pixHD that added a learning mechanism for high-resolution images in two generators. Moreover, this model learns distinct boundaries for similar objects using instance maps.

Lee et al. [66] proposed MaskGAN, which specializes in the generation of face images from face labels. Park et al. [67] proposed an improved model, spatially adaptive denormalization (SPADE), which tackled shortcomings of class information deficits in the input process of label images for batch normalization. SPADE provides various image transformations for simple input labels with flexibility in scaling and bias rates in each layer. Sushko et al. [68] proposed an improved model, with only adversarial supervision for semantic image segmentation (OASIS), which applied segmentation networks to SPADE discriminators. Although SPADE used VGGNet [69], which contains numerous parameters as a backbone for the discriminator of SPADE [67], OASIS provides a lightweight network through the introduction of a simplified regionally segmented network. Moreover, OASIS provides multimodal images with the direct input of 3D noise tensors and labels images to a discriminator, with transformed images based on multidimensional information [68].

As a representative unconditional GAN, Zhu et al. [70] proposed CycleGAN for actualizing cycle consistency. To learn the relational features between input features and output values, cycle consistency loss (CCL) was introduced to CycleGAN for reconstructing input images from output images as inverse mapping in low-dimensional feature spaces. Park et al. [71] proposed contrastive unpaired translation (CUT) based on contrastive learning (CL). They extracted simulated features such as background and object structure in input images to learn patch regions without using comprehensive learning for entire images. Based on the U-Net [72] architecture, Eskandar et al. [73] proposed an unsupervised paradigm for semantic image synthesis (USIS) consisting of a generator and a discriminator derived from SPADE [67]. This is a novel framework combined with an unconditional GAN with contrastive self-supervised learning (SSL) [74] based on image segmentation. Moreover, entered images were recognized based on wavelet transformations for matching the color and texture distributions in source images. The experimentally obtained results obtained using three public datasets demonstrated bridging of the performance gap separating paired and unpaired translation models [75].

Numerous data augmentation methods using GANs have been proposed in conjunction with academic research and industrial DL applications [76–78]. Although research on data augmentation is advancing rapidly, no meta approach has yet been proposed. Recognition accuracies are competitive and interchangeable depending on benchmark datasets. Moreover, popular benchmark datasets comprise images of general objects taken as scenes in various environments. Because sensing targets such as crops and weeds grow and change diversely over time [79], research projects and case studies using agricultural datasets are more limited than those presented for other computer vision tasks [80].

### 3. Proposed Framework

*3.1. Overall Network Architecture*

Figure 1 presents the overall network architecture of our proposed framework. This framework globally comprises three modules: GAN modules of two types and a semantic segmentation module.

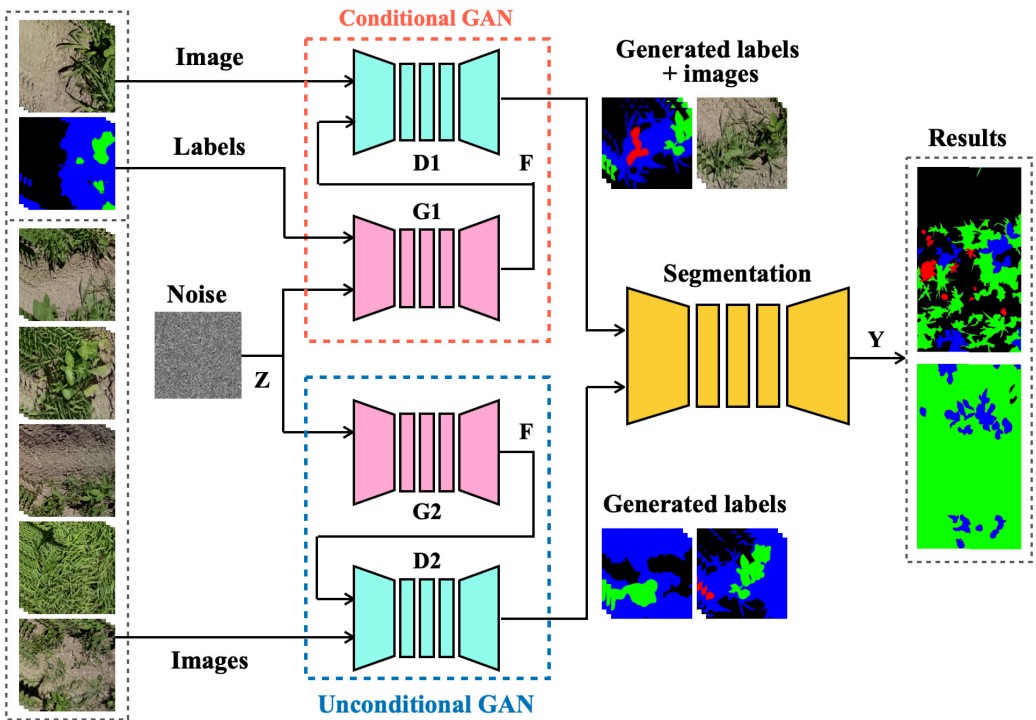

**Figure 1.** Overall network architecture of the proposed framework consisting of three modules.

For conventional semantic segmentation methods [81–83] based on supervised learning, source images and corresponding label images were prepared in advance. Training images including validation images and test images were extracted from these image sets. Our proposed framework provides image augmentation using GANs of two types. Subsequently, source and augmented images and labels were provided to the semantic segmentation module for pixel-wise segmentation. The utilization of both GAN modules provides not only data augmentation based on image generation, but also automatic labeling for the limited number of source images. The conditional GAN module is applied if a source image and its corresponding label are present. By contrast, the unconditional GAN module is applied if source images are solely present without label images.

### 3.2. GAN

Enormous computational resources must be used for GANs [84]. To assess both the extended capacity and computational cost, we used CUT [71] for an unconditional GAN and pix2pixHD [65] for a conditional GAN as the baseline for this study. Using both outstanding GANs, our framework provides not only distinctive feature learning among domains, but also automatic labeling and data augmentation based on simulated image creation. Moreover, both models are useful to achieving semi-conditional translation with multiple GANs as a future application. Figure 2 depicts structural differences between a conditional GAN and an unconditional GAN. The flows of label images given to a generator are distinctly different.

#### 3.2.1. Unconditional GAN

Based on the framework of CycleGAN [70], CUT [71] introduced contrastive representation learning mechanism [85]. Let $x$, $\hat{y}$, and $z$ respectively denote an input image, an output image, and a patch region and letting $N$ denote the number of image sampling; then, a positive $x$ sample, $z_n^+$ in $N$ and $\hat{y}$ in $Z$, is trained as a similar pair. By contrast, for a negative $x$ sample, $z_n^-$ in $N$ and $\hat{y}$ in $Z$ are trained as a non-similar pair.

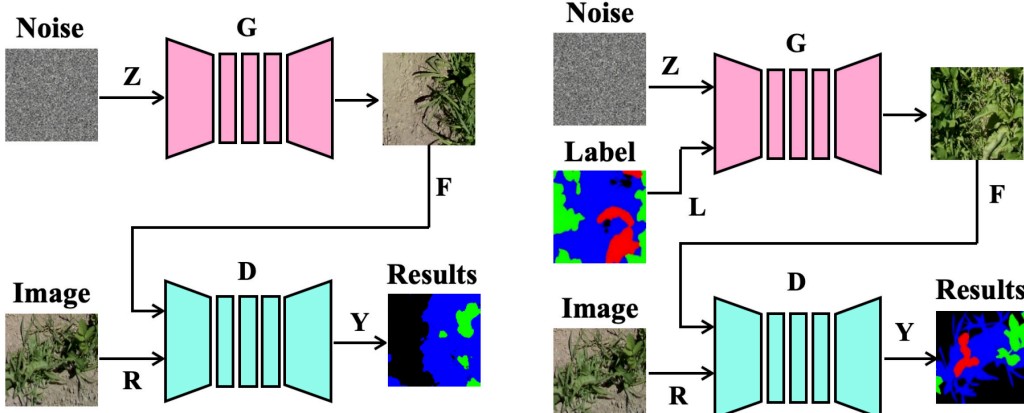

**Figure 2.** Structural differences between conditional GAN and unconditional GAN.

For a classification problem of $N + 1$ patterns, softmax cross-entropy $L_s$ [86] is calculated as

$$L_s(z, z^+, z^-) = -\log\left[\frac{\exp(z \cdot z^+ / \tau)}{\exp(z \cdot z^+ / \tau) + \sum_{n=1}^{N} \exp(z \cdot z_n^- / \tau)}\right], \tag{1}$$

where $\tau$ represents a metric correction coefficient for $z^+$ and $z_n^-$ in $z$.

To calculate the obtained patch values, $L_p$ was introduced as a loss function for a particular patch combined with the conventional loss function $L_g$ for a GAN. $L_g$ and $L_p$ are defined as

$$L_g(G, D, X, Y) = \mathbb{E}_y \log D(y) + \mathbb{E}_x \log(1 - D(G(x))), \tag{2}$$

$$L_p(G, X) = \mathbb{E}_x \sum_{s=1}^{L} \sum_{t=1}^{T} \ell(z_{s,t}, z_{s,t}^+, z_{s,t}^-), \tag{3}$$

where $X, Y, s \in S$, and $t \in T$ respectively denote an input image domain, an output image domain, an index value for sampling layers, and the sampling number in a patch region at the $s$-th layers.

After adding the Formulas (2) and (3), the final $L$ for CUT is obtained as the following.

$$\min_{G} \max_{D} L = L_g(G, D, X, Y) + L_p(G, X) + L_p(G, Y). \tag{4}$$

After learning patch regions, structural object features and the background of input images are extracted simultaneously.

### 3.2.2. Conditional GAN

To translate images with higher resolutions and quality, pix2pixHD [65] was proposed as an improved version of pix2pix [64]. The pix2pixHD network includes two subnetworks: a global generator ($G1$) and a local enhanced network ($G2$). First, $G1$ is trained using low-resolution images. After training, $G2$ is connected to $G1$. Subsequently, both networks are trained using high-resolution images. The $G1$ weights are fixed partially while the training proceeds. Finally, both networks are retained after releasing the $G1$ weights. Stability and convergence are improved based on a mechanism that allows the system to learn incrementally for images of different resolutions.

To enhance the network architecture, pix2pixHD copes with feature unevenness caused by scale changes using three discriminators. Each discriminator extracts salient features from each layer. To assess the calculation of the numerical feature matching degrees

obtained from source images and augmented images, feature matching loss $L_m$ based on the Manhattan metric in the taxicab geometry [87] is defined as

$$L_m(G, D_k) = \mathbb{E}_{(x,y)} \sum_{t=1}^{T} \frac{1}{N_t} \parallel D_k^t(x, y) - D_k^t(x, G(x)) \parallel, \tag{5}$$

where $D_k$ and $T$ respectively represent the $k$-th discriminator and the number of layers in each discriminator.

After adding Equations (2) and (5), the final loss $L$ of pix2pixHD is presented as

$$\min_{G} \max_{D} L = \sum_{k=1}^{K} L_g(G, D_k, X, Y) + \lambda \sum_{k=1}^{K} L_m(G, D_k), \tag{6}$$

where $\lambda$ and $K$ respectively stand for a weight coefficient of $L_m$ and the number of discriminators.

### 3.3. Semantic Segmentation

Semantic segmentation provides object and stuff class labels for every pixel in a presented image. This process is also known as a high-density labeling task [88]. Images include widely diverse explicit and implicit information and meanings. Moreover, semantic segmentation provides a comprehensive scene description that includes information related to object category, location, shape, and context [89]. Semantic segmentation has numerous applications such as medical image processing [90], self-driving cars [91], structural damage mapping [92], vehicle detection [93], remote sensing [94], monitoring photovoltaic solar plants [95], power line cable inspection [96], building extraction [97], drone detection [98], classifying cooking activities [99], socially assistive robots [100], and industrial automation [101]. Representative semantic segmentation models include SegNet [102], U-Net [72], DeepLabV3+ [103], HRNet [104], SegFormer [41], Segmenter [42], and ConvNeXt [105].

For visual tasks, vision transformer (ViT) [106] models perform better than ConvNets [4] if segmentation targets are objects [107]. By contrast, for stuff classes, especially in textures, the convolution mechanism provided by ConvNets demonstrated superiority, including in feature representations based on multilayered relations, compared to attention mechanisms provided by ViT [108]. Steiner et al. [109] indicated that the weaker ViT inductive bias is generally found to cause an increased reliance on model regularization or data augmentation when training on smaller training subsets. Therefore, for extracting sufficient ViT performance, very large datasets such as ImageNet-21K [110] or JFT-300M [111] are necessary for pre-training [112].

Considering a balance between classification accuracy, implementation platform, and processing efficiency, DeepLabV3+ [103] was used for this study. This network flexibly extracts features with parallel processing of multiple filters that contain arbitrary resolutions by spatial pyramid pooling. The automatic formation of a decoding structure that propagates spatial information held by an encoder provides extraction of clear boundaries between objects. DeepLabV3+ [103] was pre-trained using the COCO-Stuff [113] open benchmark dataset, which provided comprehensive annotations of 172 classes. Based on a comparison of results obtained from our earlier studies [114,115], Xception-65 [116] was used for the network backbone in this study. Here, the semantic segmentation module can be switched flexibly to state-of-the-art or practical backbones according to technological progress, target problems, and specifications, such as ConvNeXt [105], Segmenter [42], SegFormer [41], TransFG [117], MetaFormer [118], and K-Net [119].

## 4. Requirements

*4.1. Benchmark Datasets*

For this study, we used three benchmark datasets. The first benchmark dataset is Cityscapes [120], which is a public dataset for semantic urban scene understanding. The second and third datasets are our originally developed datasets: Rice blast, which is a rice disease detection dataset developed for our earlier study [80], and Soybeans, which is a leaf and weed classification dataset developed for this study.

### 4.1.1. Cityscapes

Cityscapes [120] is a benchmark suite for training and validating pixel-wise and instance-level semantic labeling models. Based on this pioneering work, derivative datasets have been proposed such as Citypersons [121], CityWalks [122], and Cityscapes 3D [123]. The Cityscapes suite consists of a large and diverse set of stereo video sequences recorded on the streets of 50 German cities. All images converted from the video sequences were annotated as 20 classes for semantic labels and eight classes for instance labels. For this study, we used images with semantic labels because of the similar specifications in our original datasets.

Figure 3 depicts sample images and respective annotations as GT. The semantic labels and corresponding colors are presented in Table 1. The original image resolution is 2048 × 1024 pixels. The region of interest (RoI) sizes for random sampling are 512 × 256 pixels for the semantic segmentation module and 256 × 256 pixels for the GAN modules. We set these RoI sizes considering the computation time and memory capacity in our implementation environment. We split 3000 randomly extracted images in a 4:1:1 ratio for training, validation, and test subsets.

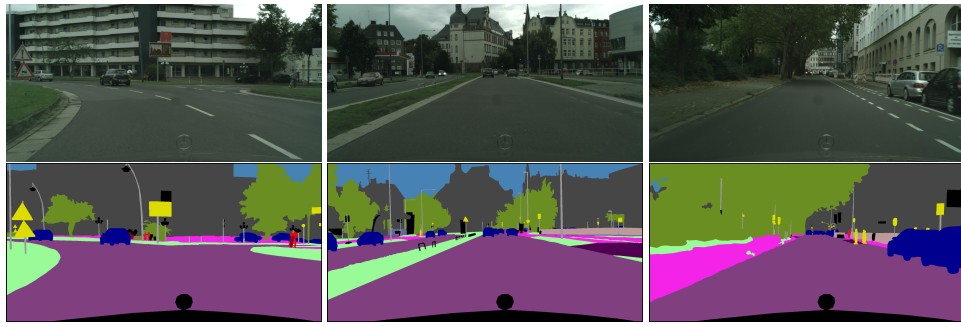

**Figure 3.** Sample images of Cityscapes: source images (**upper**) and GT images (**lower**).

**Table 1.** Semantic labels and corresponding colors.

| ID | Class | Color | ID | Class | Color | ID | Class | Color |
|----|-------|-------|----|-------|-------|----|-------|-------|
| 0 | unlabeled | ⬛ | 1 | ground | 🟪 | 2 | road | 🟪 |
| 3 | sidewalk | 🟪 | 4 | building | ⬛ | 5 | wall | 🟦 |
| 6 | fence | 🟫 | 7 | pole | ⬜ | 8 | traffic light | 🟧 |
| 9 | traffic sign | 🟨 | 10 | vegetation | 🟩 | 11 | terrain | 🟩 |
| 12 | sky | 🟦 | 13 | person | 🟥 | 14 | rider | 🟥 |
| 15 | car | 🟦 | 16 | truck | ⬛ | 17 | bus | 🟦 |
| 18 | train | 🟦 | 19 | motorcycle | 🟦 | 20 | bicycle | 🟥 |

### 4.1.2. Rice Blast

For the second benchmark dataset, we used this benchmark dataset from our earlier work [80]. Figure 4 depicts sample images and corresponding labels after annotation. This dataset comprises three classes: background (BG), rice leaves (RL), and rice blast (RB). The black, green, and red color pixels respectively correspond to BG, RL, and RB classes. This

dataset comprises 119 images with a resolution of 6240 × 4160 pixels. Random sampling RoI sizes are 480 × 320 pixels for the semantic segmentation module and 256 × 256 pixels for the GAN modules. Images were split according to a 7:2:3 ratio for training, validation, and test subsets.

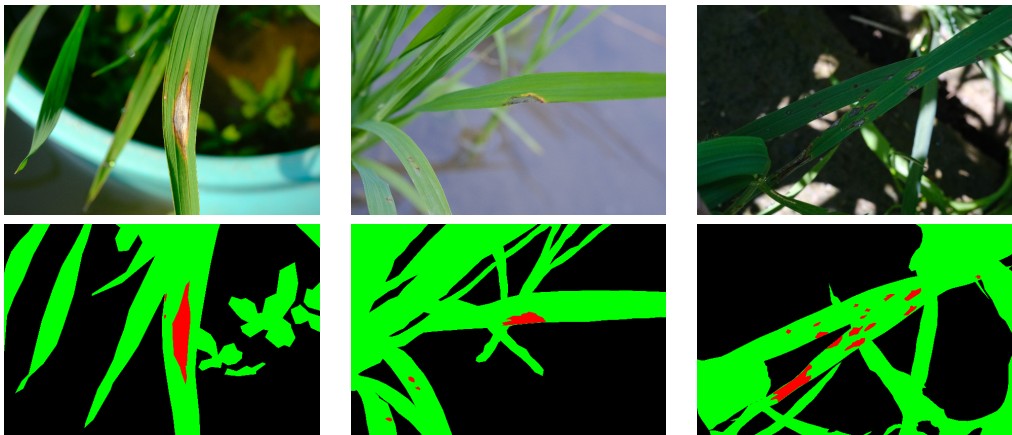

**Figure 4.** Sample images of Rice blast: source images (**upper**) and GT images (**lower**).

### 4.1.3. Soybeans

For this study, we created a new benchmark dataset designated as Soybeans. Figure 5 depicts sample images and corresponding labels after annotation. This dataset comprises four classes: BG, soybean leaves (SL), Poaceae weeds (PW), and other weeds (OW). The black, green, red, and blue color pixels respectively correspond to BG, SL, PW, and OW classes. We obtained an HD movie using an onboard camera mounted on a drone (Phantom 4; SZ DJI Technology Co., Ltd., Shenzhen, China). We extracted 300 images with resolution of 1280 × 720 pixels from the movie. The RoI sizes for random sampling were set as 512 × 256 pixels for the semantic segmentation module and 480 × 360 pixels for the GAN modules. The images were split according to a 4:1:1 ratio for training, validation, and test subsets.

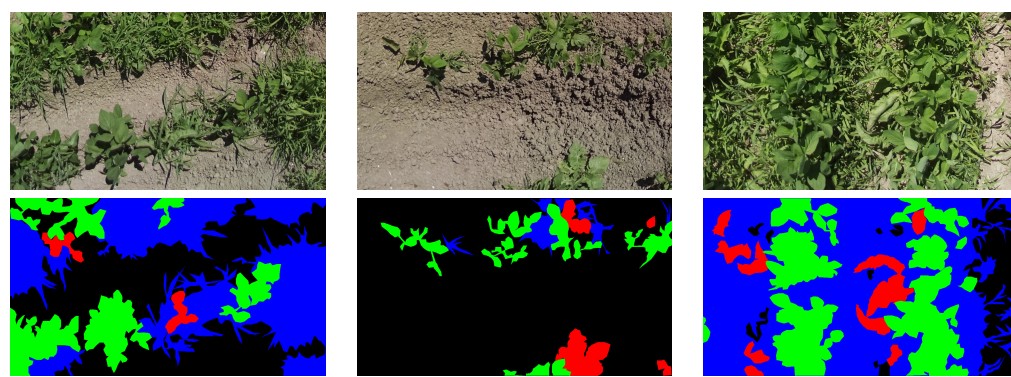

**Figure 5.** Sample images of Soybeans: source images (**upper**) and GT images (**lower**).

### 4.2. Evaluation Metrics

We used the Fréchet Inception distance (FID) [124] to assess the quality of images generated using GANs. FID is an evaluation criterion based on the Inception [33] network backbone, which was pre-trained using ImageNet [125]. Similarities are evaluated quantitatively from feature vectors extracted from source images and augmented images. Letting $m_w$ and $m$ respectively denote norms of feature vectors obtained from augmented images

and source images, and letting $C_w$ and $C$ respectively denote the respective feature vector and covariance matrixes, the definition of FID can be presented as

$$\text{FID} = \parallel m - m_w \parallel^2 + Tr(C + C_w - 2\sqrt{CC_w}), \tag{7}$$

where $Tr$ stands for a trace of matrixes.

Subsequently, we used intersection over union (IoU), which is also known as the Jaccard index [126], as a semantic segmentation evaluation metric. IoU is the de facto evaluation metric used for object challenges such as PASCAL VOC [127] and MS COCO [128]. Using IoU, the degree of overlapping in the GT regions and prediction regions is defined as

$$\text{IoU} = \frac{TP}{TP + FP + FN}, \tag{8}$$

where $TP$, $FP$, and $FN$ respectively denote true positive, false positive, and false negative rates.

## 5. Preliminary Experiment

Our proposed method generates labels from images and images from labels using CUT [71] as an unconditional GAN and pix2pixHD [65] as a conditional GAN. This preliminary experiment provides accuracy and characteristics of labels generated from images and images generated from labels. For this study, the processes to be generated from images to labels and labels to images are represented as I2L and L2I.

### 5.1. Cityscapes

Figure 6 depicts time-series training loss trends for the respective GANs in Cityscapes. For the trends of CUT, the I2L loss from the discriminator was lower than that from the generator after 150 epochs. By contrast, the L2I losses from the generators and the discriminators of both GAN modules are competitive, respectively, at around 0.4 and 0.6.

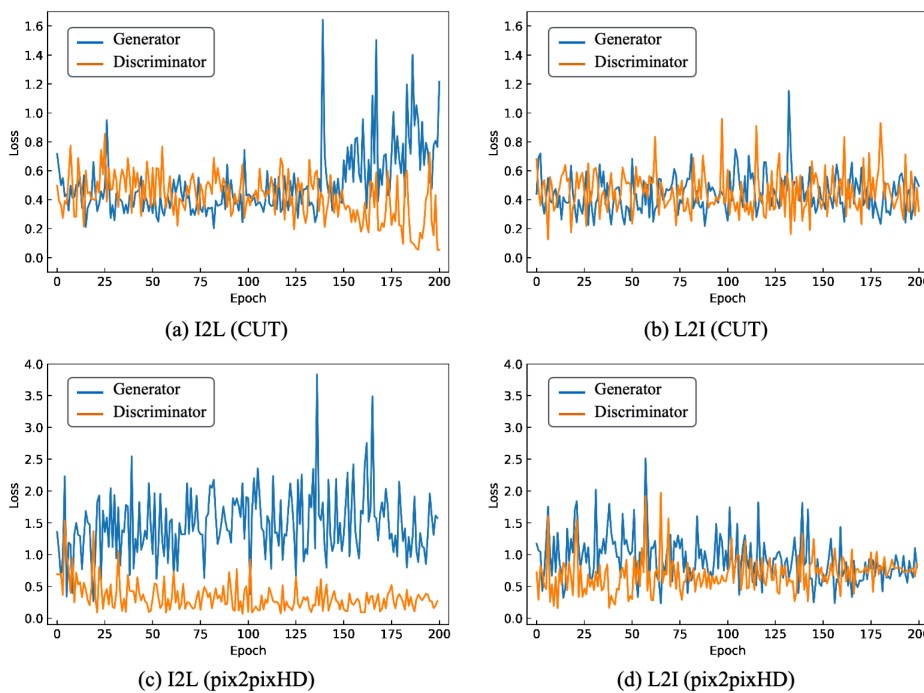

**Figure 6.** Time-series training loss trends for the respective GANs in Cityscapes.

Figure 7 depicts FID trends at 25 epoch intervals in respective GANs. The FID scores, except for I2L for CUT, dropped according to the progress of epochs. As the minimum, the

L2I FID score for CUT showed 27.6 at the final epoch. By contrast, the I2L FID for CUT is unchanged after increasing the score from 75 epoch to 100 epoch. This result demonstrates the difficulty of the generation process from images to labels, with CUT being harder than the other three generators.

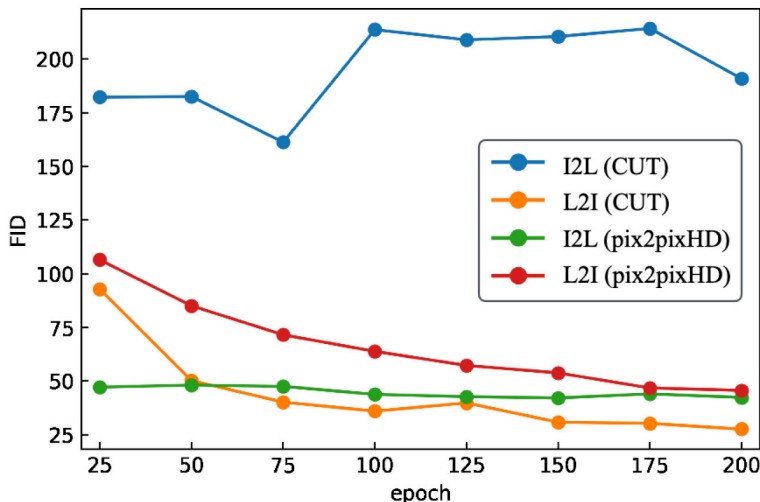

**Figure 7.** Time-series FID trends for the respective GANs in Cityscapes.

Figure 8 depicts sample images of style transfer results. These results were obtained from the final generation of 200 epochs except for the I2L result for CUT. For the FID presented in Figure 7, the result shown in Figure 8a was obtained at the generation of 175 epochs. The first and third columns respectively present the source images and GT images. The second and fourth columns respectively present I2L results and L2I results. Therefore, label generation results can be compared subjectively in the images of the second and third columns. Moreover, augmented images can be compared subjectively to the images in the first and fourth columns. Two sets of results are shown for each transfer and each GAN. The first and third rows present transfer results of high FID scores. The second and fourth rows present transfer results of low FID scores.

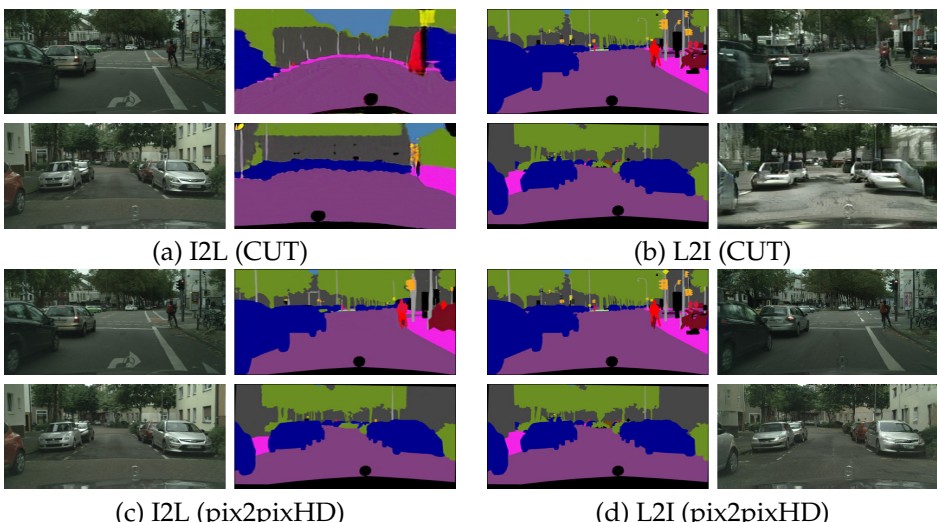

(a) I2L (CUT)  (b) L2I (CUT)

(c) I2L (pix2pixHD)  (d) L2I (pix2pixHD)

**Figure 8.** I2L and L2I style transfer results for the respective GANs in Cityscapes.

Although pixels of the source images and the label image are widely matched in stuff in terms of roads and vegetation as stuff for I2L of CUT in Figure 8a, the objects in terms of cars and riders are unmatched. By contrast, the I2L results of pix2pixHD in Figure 8c

ensure high reproducibility for both stuff and objects. The L2I result for CUT in Figure 8b demonstrates that distortion occurred throughout the images. Although cars and riders were generated as objects, discriminating them is difficult without consideration of the traffic scene context. The L2I result for pix2pixHD in Figure 8d demonstrates that an image resembling GT was generated. Although small differences are apparent, such as color patterns, road signs painted on the road surface, traffic dashes and solid lines, and overall shapes of the windows and doors of the buildings, this result can be evaluated as having sufficient quality, similar to an actual image. As depicted in Figure 6, the pix2pixHD loss is lower than the CUT loss. As depicted in Figure 7, the FID of pix2pixHD is lower than that of CUT. Visual observation-based evaluation results verify that the incongruity of pix2pixHD is lower than that of CUT. From these results, we infer that the quality of generated labels and images for pix2pixHD is higher than those for CUT. We consider that this trend corresponds to the property of the objective evaluation metrics of loss and FID.

### 5.2. Rice Blast

Figure 9 depicts time-series training loss trends for the respective GANs in Rice blast. Although the L2I losses for CUT and the I2L losses for pix2pixHD are mutually close, the I2L losses for CUT are separated. For I2L of CUT and L2I of pix2pixHD, the discriminator loss is lower than the generator loss. However, the I2L losses for CUT are mutually close.

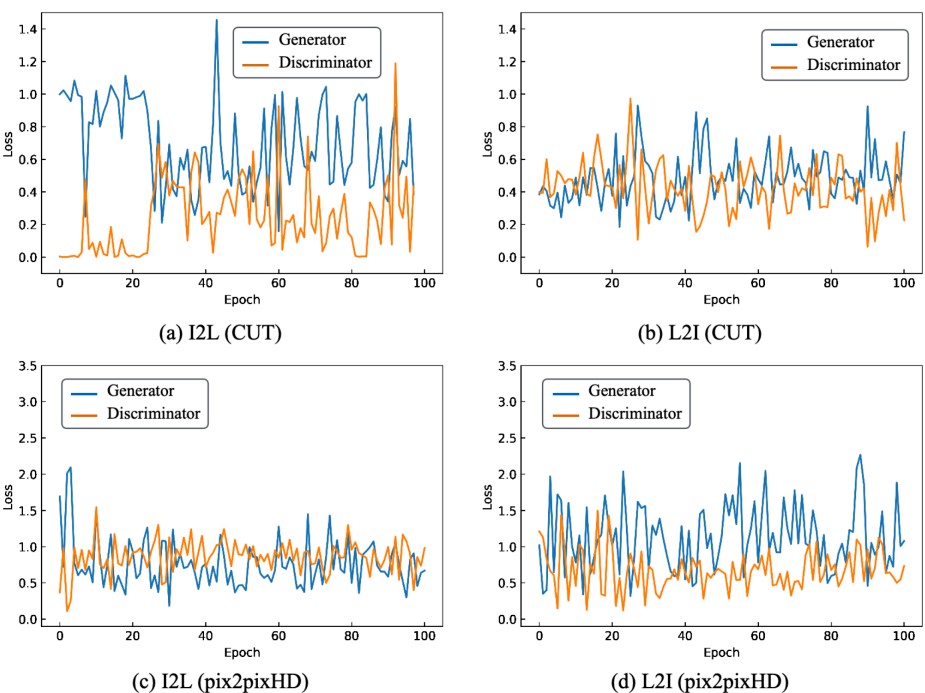

**Figure 9.** Time-series training loss trends for respective GANs in Rice blast.

Figure 10 depicts time-series FID trends at 25 epoch intervals in the respective GANs. The L2I FID scores for pix2pixHD dropped smoothly according to the forwarded generations. The FID at 100 epoch is 205.1. By contrast, FID scores of the other three models showed no change according to the increased generations. The I2L FID for pix2pixHD was 170.9 at 100 epoch. This score is low compared to that of other models. The L2I FID for CUT shows two-times higher results than those obtained with other models. This tendency indicates difficulties in ensuring accuracy compared with other models, especially in the L2I process for CUT.

Figure 11 depicts sample images of style transfer results. These results were obtained from the final generation of 100 epochs. Unmatched labels appeared in label images of I2L for CUT, especially in RB regions. The L2I result for CUT demonstrates that object boundaries were created according to the boundaries of labels; the augmented images are

unnatural. For I2L results for pix2pixHD, the RB positions are correct, but the shapes and sizes are unmatched. The L2I results for pix2pixHD show the occurrence of shadow regions that dropped the contrast of RL regions partially. Although RB regions lack clarity, virtual blasts similar to those in the source images were generated.

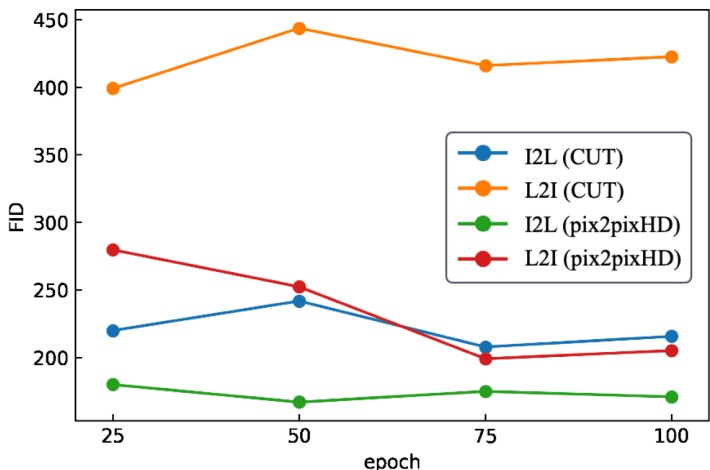

**Figure 10.** Time-series FID trends for the respective GANs in Rice blast.

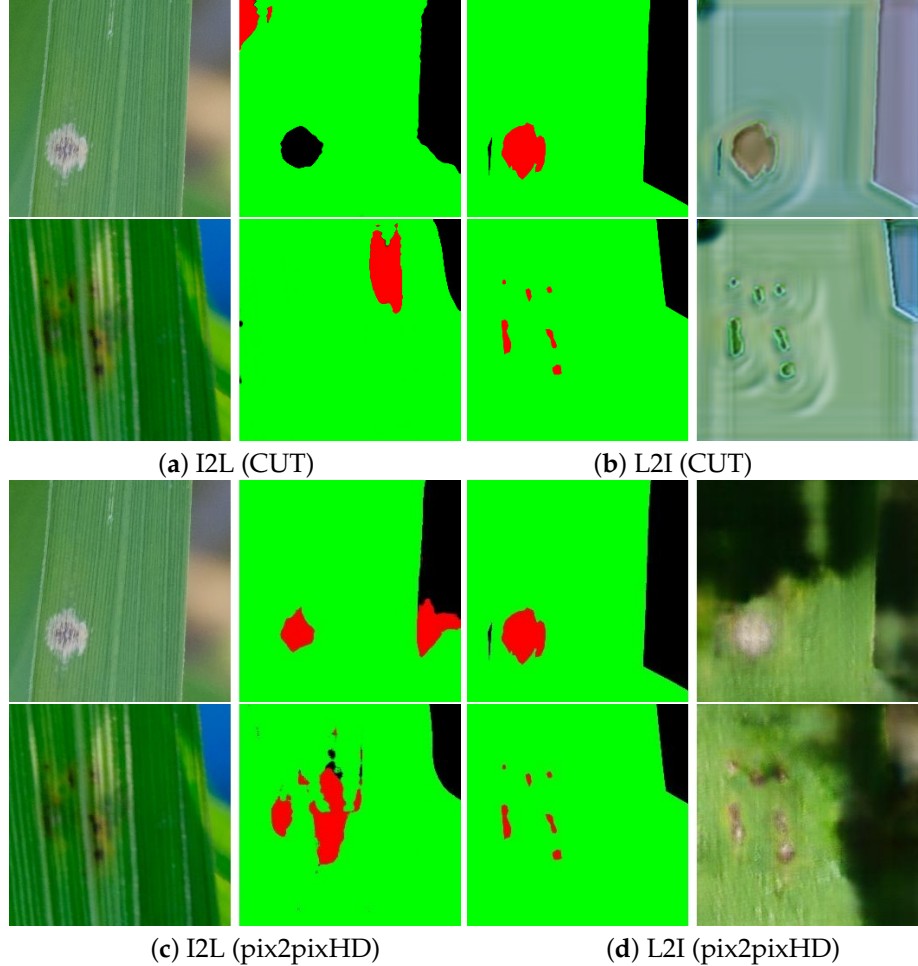

**Figure 11.** I2L and L2I style transfer results for the respective GANs in Rice blast.

### 5.3. Soybeans

Figure 12 presents time-series training loss trends for respective GANs in Soybeans. The pix2pixHD result demonstrates similar I2L loss trends in the generator and the discriminator. Results for the other three models indicate that the discriminator losses are lower than the generator losses.

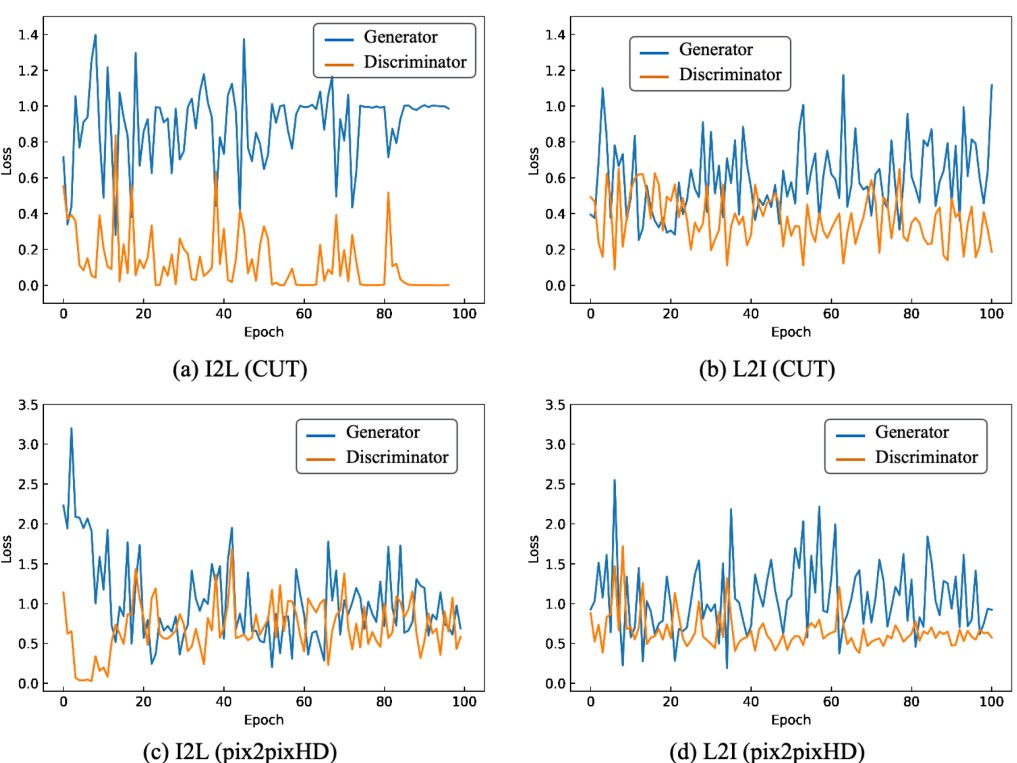

(a) I2L (CUT)

(b) L2I (CUT)

(c) I2L (pix2pixHD)

(d) L2I (pix2pixHD)

**Figure 12.** Time-series training loss trends for the respective GANs in Soybeans.

Figure 13 portrays time-series FID trends at 25 epoch intervals in the respective GANs. Overall, the L2I FID scores tended to decrease as the generations progressed. The I2L FID scores for pix2pixHD showed no change through generations. The I2L FID scores for CUT increased after 50 generations.

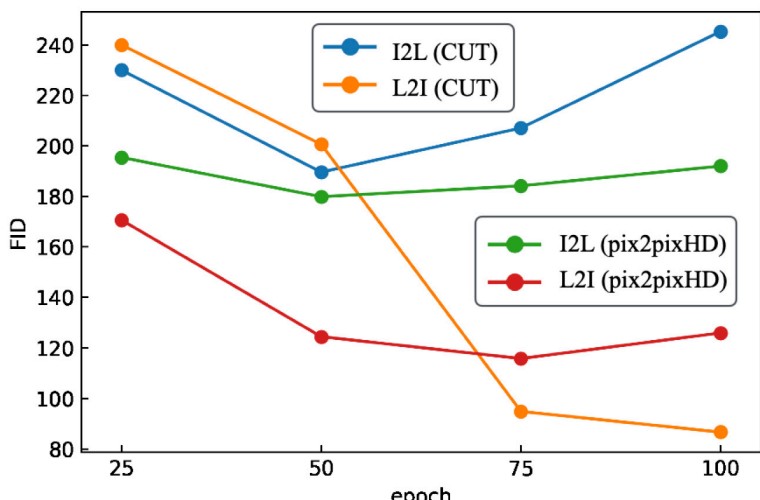

**Figure 13.** Time-series FID trends for the respective GANs in Soybeans.

Figure 14 depicts sample images of style transfer results. These results were obtained from the final generation of 100 epochs. The I2L results for CUT include labels that were converted incorrectly from BG to SL. The L2I results for CUT include numerous pixels that were converted incorrectly from OW to BG. The I2L results for pix2pixHD show labels generated according to GT with slight shape differences. The L2I results for pix2pixHD show no false transformation pixels that differ from those for CUT. Compared to the source images, transferred images were generated with high reality with only slight differences in leaf shape and sharpness.

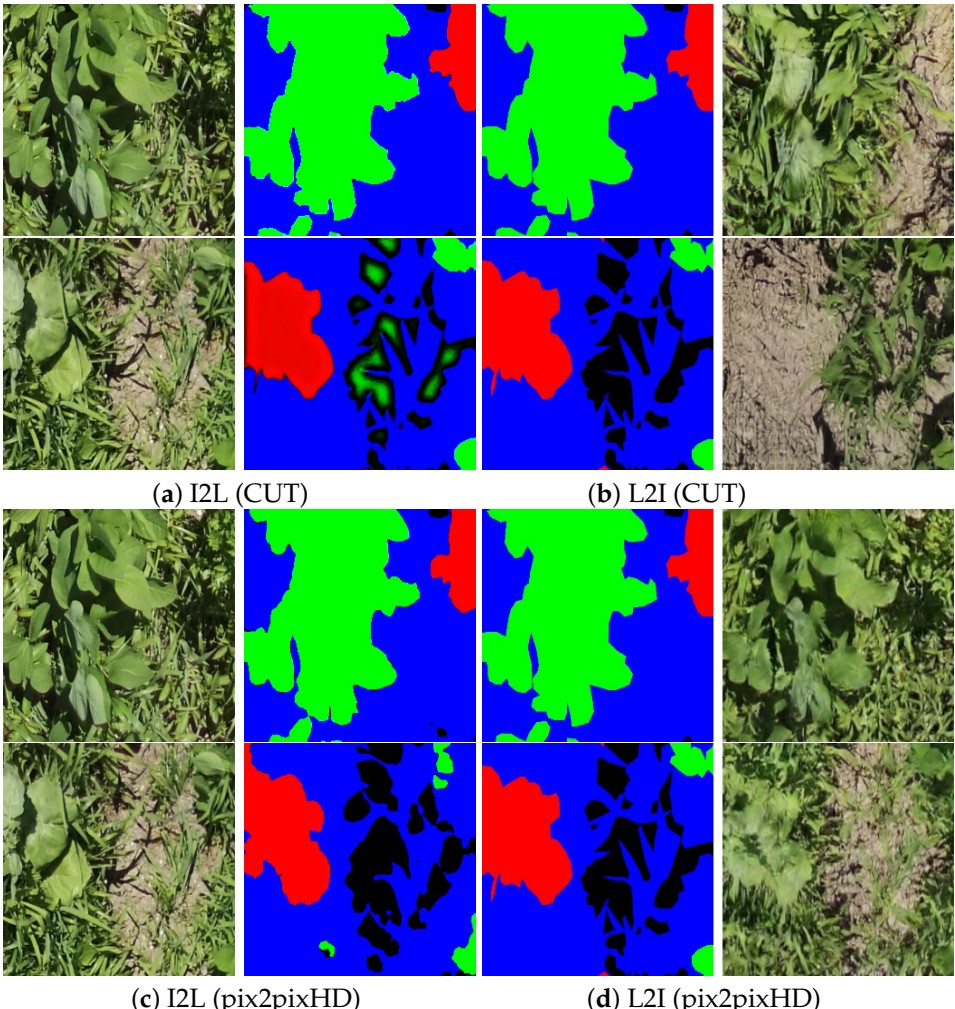

(**a**) I2L (CUT)    (**b**) L2I (CUT)

(**c**) I2L (pix2pixHD)    (**d**) L2I (pix2pixHD)

**Figure 14.** I2L and L2I style transfer results for the respective GANs in Soybeans.

## 6. Evaluation Experiment

### 6.1. Setup

For this study, we evaluated the proposed framework quantitatively and qualitatively using five experiments, including baseline evaluation. Table 2 presents a summary of the data source combinations for each experiment. First, semantic segmentation was conducted using only the source images. We considered this result as the baseline. Subsequently, Experiment A was conducted using images generated from CUT [71] as an unconditional GAN. Experiment B was conducted using images generated from pix2pixHD [65] as a conditional GAN. Experiment C was conducted using images generated from both CUT and pix2pixHD. Finally, Experiment D was conducted using all the images.

**Table 2.** Data combination for each experiment.

| Data Source | Baseline | Exp. A | Exp. B | Exp. C | Exp. D |
|---|---|---|---|---|---|
| Source images | ✓ | | | | ✓ |
| CUT [71] | | ✓ | | ✓ | ✓ |
| pix2pixHD [65] | | | ✓ | ✓ | ✓ |

Table 3 presents hyperparameters of each model for the experiments. The CUT and pix2pixHD parameters were set through the literature [65,71] and our preliminary experiments. The DeepLabV3+ parameters were set empirically through our earlier studies [114,115] and several references [129–131]. These settings were not necessarily optimal values because we used no automated parameter setting tools [132–134].

**Table 3.** Hyperparameters and setting values.

| Model | CUT [71] | pix2pixHD [65] | DeepLabV3+ [103] |
|---|---|---|---|
| Backbone | ResNet-50 [34] | VGG-19 [69] | Xeception-65 [116] |
| Training iterations [epoch] | 200 | 300 | |
| Training coefficient | 0.0002 | 0.0010 | |
| Mini-batch size | 4 | 2 | |
| Sampling size [pixel] | $256 \times 256$ | $480 \times 360$ | |
| Number of sampling | 2500 | 2000 | |

We used two platforms for implementing and executing our framework. First, we used Colaboratory (Colab) (Google LLC; Mountain View, CA, USA), which is a cloud platform for accelerating DL applications [135]. The major Colab specifications are the following: 12 GB Tesla K80 (NVIDIA Corp., Santa Clara, CA, USA), GPU; XEON 2.20 GHz (Intel Corp., Santa Clara, CA, USA), CPU; Ubuntu 18.04.5 LTS, OS; 12.69 GB, RAM; 107.72 GB, HDD for CPU and TPU; and 78.19 GB, HDD for GPU. The second platform was a Windows desktop computer. The major specifications of this platform are the following: 48 GB RTX A6000 (NVIDIA Corp., Santa Clara, CA, USA), GPU; Ryzen Threadripper 3970X 3.7 GHz (Advanced Micro Devices, Inc., Santa Clara, CA, USA), CPU; Ubuntu 18.04.5 LTS on Windows Subsystem for Linux 2 (Microsoft Corp., Redmond, WA, USA), OS; 32 GB, RAM; and 512 GB, HDD.

Although numerous GAN and semantic segmentation models have been proposed, a tradeoff exists between the performance and computational cost. For example, StyleGAN2 [136] and StyleGAN3 [137] were calculated using eight GPUs (Tesla V100, NVIDIA Corp., Santa Clara, CA, USA). For our implementation and execution environment, we used the models and parameters presented in Table 3. These models can be changed arbitrarily according to the computational platform because the proposed method is the framework depicted in Figure 1.

### 6.2. Baseline

As a baseline for this study, we trained DeepLabV3+ using only the source images without augmented images generated from GANs.

### 6.2.1. Cityscapes

Figure 15 depicts the confusion matrix for the Cityscapes test subset images. After normalization in each class, each matrix element is displayed in heatmap color distributions. The diagonal color pattern from the upper left to the bottom right represents high temperature. However, the color patterns for IDs 7, 8, 9, 14, 18, and 19, which correspond to `pole`, `traffic light`, `traffic sign`, `rider`, `train`, and `motorcycle`, show high temperatures in the other classes of the non-diagonal positions.

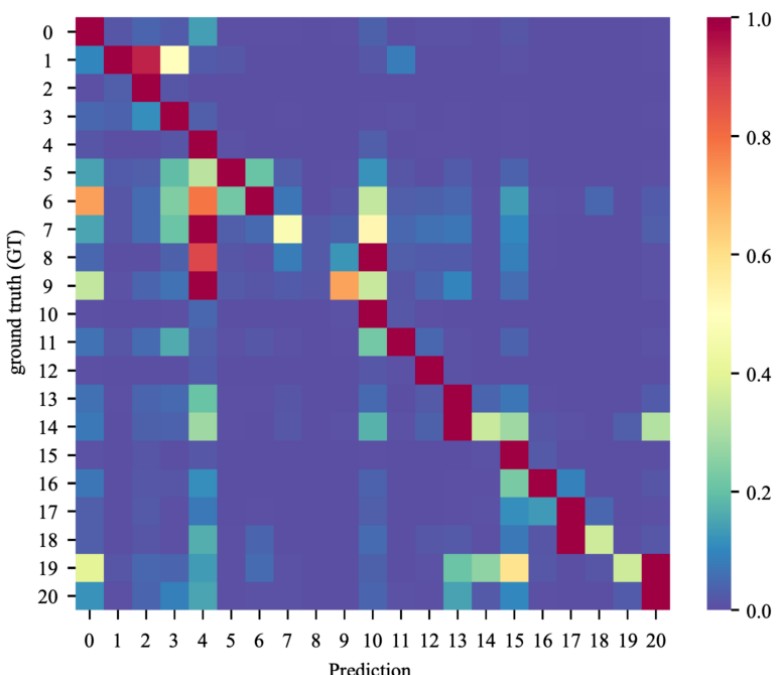

**Figure 15.** Confusion matrix for Cityscapes test subset images.

Table 4 presents IoU scores of the semantic segmentation result in each class of Cityscapes. The corresponding class IDs are apparent in Table 1. For the training subsets, the IoU scores of 10 classes, which respectively correspond to `unlabeled`, `road`, `sidewalk`, `building`, `vegetation`, `terrain`, `sky`, `car`, `bus`, and `train`, are greater than the mean score. For the test subsets, the IoU scores of eight classes, which respectively correspond to `unlabeled`, `ground`, `road`, `sidewalk`, `building`, `vegetation`, `sky`, and `car`, are greater than the mean score.

**Table 4.** IoU scores for the respective Cityscapes classes.

| ID | 0 | 1 | 2 | 3 | 4 | 5 | 6 | 7 | 8 | 9 | 10 |
|---|---|---|---|---|---|---|---|---|---|---|---|
| Training | 0.66 | 0.20 | 0.93 | 0.66 | 0.80 | 0.29 | 0.29 | 0.13 | 0.01 | 0.25 | 0.82 |
| Test | 0.69 | 0.03 | 0.88 | 0.48 | 0.70 | 0.04 | 0.04 | 0.09 | 0.00 | 0.11 | 0.74 |

| ID | 11 | 12 | 13 | 14 | 15 | 16 | 17 | 18 | 19 | 20 | Mean |
|---|---|---|---|---|---|---|---|---|---|---|---|
| Training | 0.51 | 0.87 | 0.50 | 0.17 | 0.84 | 0.34 | 0.48 | 0.71 | 0.27 | 0.40 | 0.48 |
| Test | 0.01 | 0.64 | 0.16 | 0.02 | 0.66 | 0.03 | 0.06 | 0.01 | 0.01 | 0.16 | 0.27 |

Figures 16 and 17 respectively denote representative successful and failed test subset images. The mIoU threshold to classify these cases was set as 0.266. Figure 17 depicts false segmentation samples in the *car*, *bus*, and *truck* classes. We infer that these similar classes have difficulty learning differences among object features.

### 6.2.2. Rice Blast

Table 5 represents the confusion matrix for the Rice blast test subset images. Numbers in the parentheses correspond to recognition rates on the diagonal cells and the false recognition rates on the other cells. Although high accuracy was obtained in RL, the accuracy was low in RB. The false recognition between RB and RL is greater than in other combinations of classes.

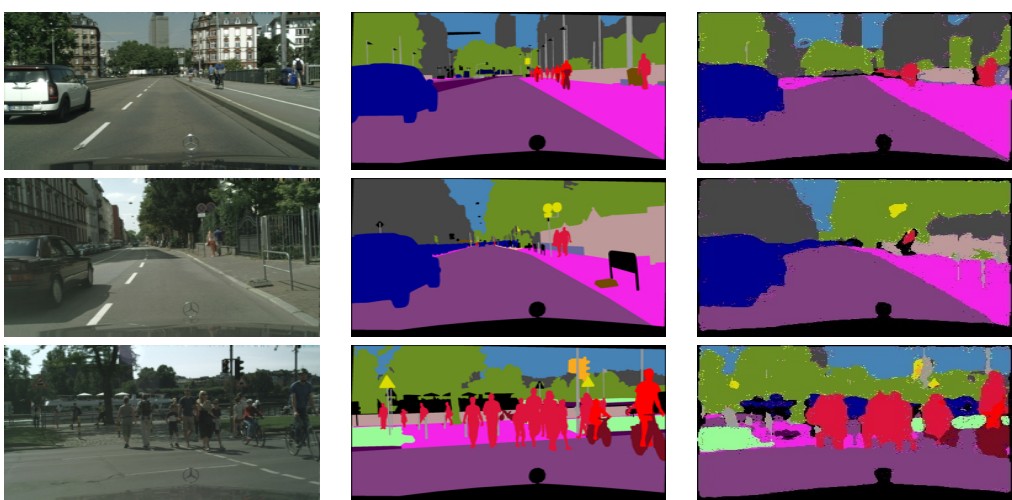

**Figure 16.** Representative successful Cityscapes images: source images (**left**), GT (**center**), and results (**right**).

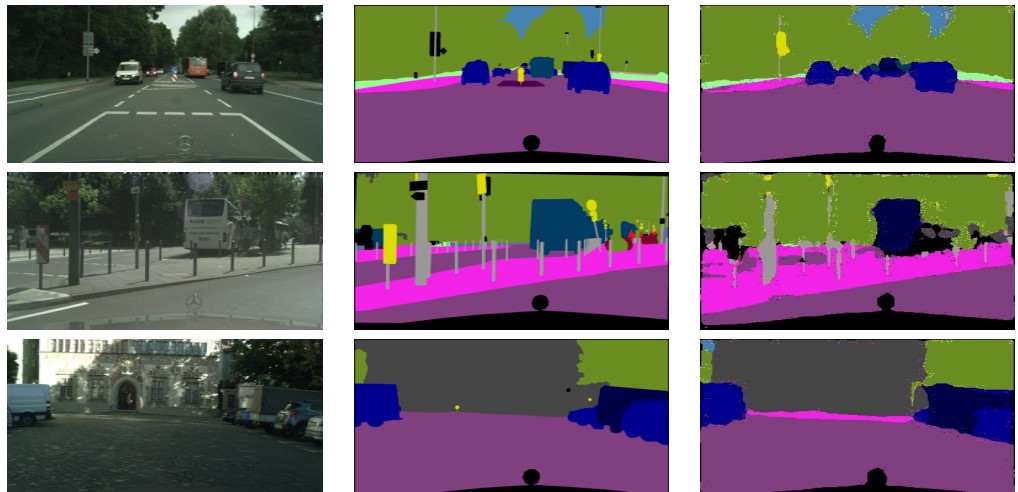

**Figure 17.** Representative failed Cityscapes images: source images (**left**), GT (**center**), and results (**right**).

**Table 5.** Confusion matrix for Rice blast test subset images [pixels].

|  | BG | RL | RB |
|---|---|---|---|
| BG | 43,544,288<br>(57.06%) | 32,753,681<br>(42.92%) | 14,875<br>(0.02%) |
| RL | 2,926,683<br>(3.17%) | 89,268,832<br>(96.79%) | 38,594<br>(0.04%) |
| RB | 26,151<br>(14.33%) | 127,392<br>(69.83%) | 28,898<br>(15.84%) |

Table 6 presents the semantic segmentation results evaluated using IoU scores for the respective Rice blast classes. The respective mIoU scores for the training and test subset images are 0.87 and 0.45. The respective highest IoU scores for the training and test RL images are 0.94 and 0.72. By contrast, the lowest IoU scores for the training and test RB images are, respectively, 0.77 and 0.10. The IoU scores of RB for the training and test subset images are, respectively, 0.10 and 0.35 lower than the mean score. We consider that the low IoU score for RB is attributable to its lower frequency of occurrence than other classes.

**Table 6.** IoU scores for the respective Rice blast classes

|          | BG   | RL   | RB   | Mean |
|----------|------|------|------|------|
| Training | 0.91 | 0.94 | 0.77 | 0.87 |
| Test     | 0.52 | 0.72 | 0.10 | 0.45 |

Figures 18 and 19 respectively present representative successful and failed test subset images. The mIoU threshold to classify these cases was set as 0.10. As the depicted images in Figure 18, RB pixels were roughly segmented from the RL regions. This result indicates the relation that RB pixels are apparent in RL regions. By contrast, as depicted images in Figure 19, false segmentation pixels exist in small RB regions with low segmentation accuracy. False segmentation pixels are apparent in dead leaf tips and RB pixels that are segmented as BG pixels. This finding from experimentation demonstrates the difficulty of learning object boundaries between RB and RL pixels.

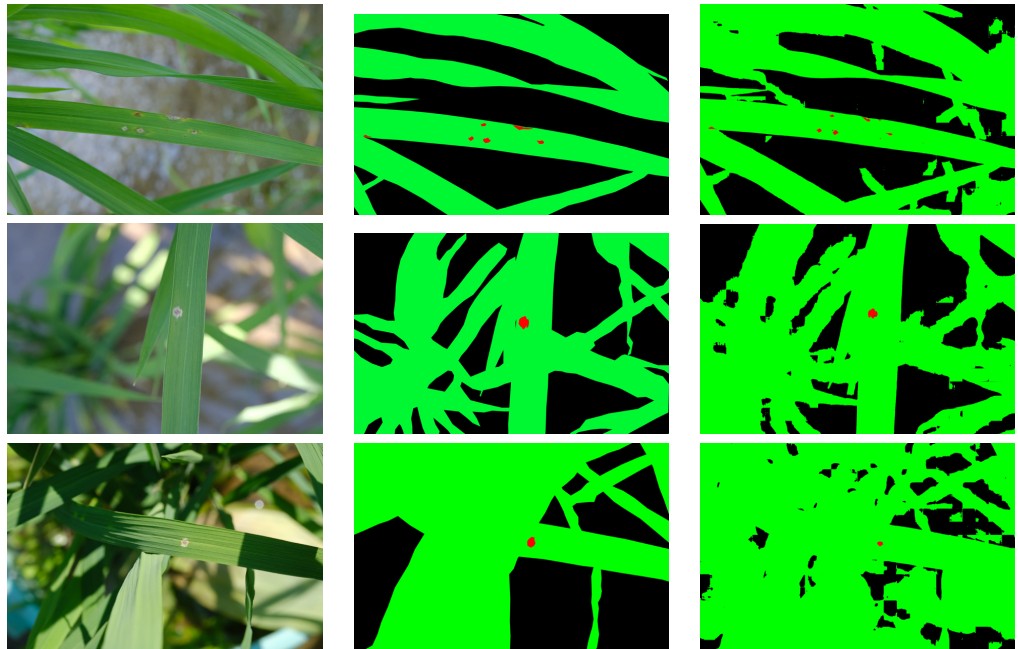

**Figure 18.** Representative successful Rice blast images: source images (**left**), GT (**center**), and results (**right**).

6.2.3. Soybeans

Table 7 represents a confusion matrix for the Soybeans test subset images. The numbers of false segmentation pixels are greater between PW and SL, and between PW and BG. However, the false segmentation rate is as high as 3% because the number of PW pixels is approximately 60% in this dataset.

Table 8 presents the semantic segmentation results evaluated using IoU scores for the respective Soybeans classes. The mIoU scores for the training and test subset images are, respectively, 0.59 and 0.56. The gap score between them is only 0.03, which is approximately a tenth of that for Rice blast. The highest IoU scores are for the training and test PW images at 0.78 and 0.83, respectively. By contrast, the lowest IoU scores are for the training and test OW images at 0.12 and 0.32, respectively. The IoU scores for the training and test OW images are, respectively, 0.47 and 0.37 lower than the mean score. We infer that this tendency is related to the OW data amount, which is merely 2% of the total data amount.

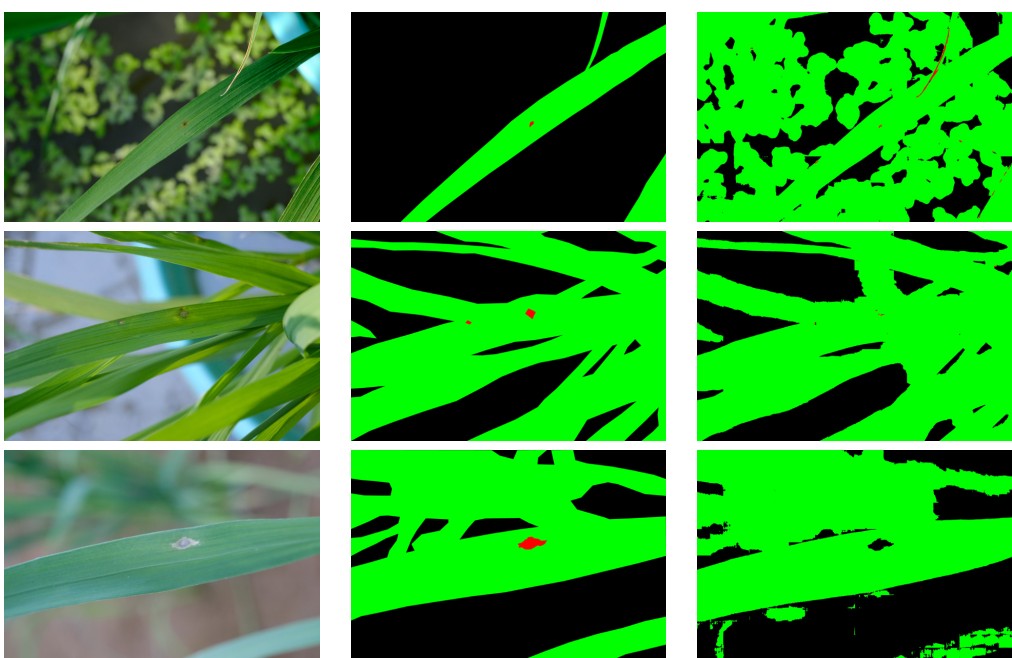

**Figure 19.** Representative failed Rice blast images: source images (**left**), GT (**center**), and results (**right**).

**Table 7.** Confusion matrix for Soybeans test subset images [pixels].

|  | **BG** | **SL** | **PW** | **OW** |
|---|---|---|---|---|
| BG | 383,495 (91.49%) | 9052 (2.16%) | 23,463 (5.60%) | 3159 (0.75%) |
| SL | 9767 (4.45%) | 162,277 (74.00%) | 39,903 (18.20%) | 7350 (3.35%) |
| PW | 50,333 (4.28%) | 37,194 (3.17%) | 1,069,904 (91.08%) | 17,227 (1.47%) |
| OW | 1109 (3.90%) | 6206 (21.81%) | 9652 (33.93%) | 11,482 (40.36%) |

**Table 8.** IoU scores for the respective Soybeans classes

|  | **BG** | **SL** | **PW** | **OW** | **Mean** |
|---|---|---|---|---|---|
| Training | 0.70 | 0.75 | 0.78 | 0.12 | 0.59 |
| Test | 0.52 | 0.56 | 0.83 | 0.32 | 0.56 |

Figures 20 and 21 respectively portray representative successful and failed test subset images. Resultant images in Figure 20 depict noise caused by unsteady segmentation at the boundaries between BG and OW pixels. Resultant images in Figure 21 show false segmentation pixels between SL and OW. Although the PW pixels are segmented correctly, several false segmentation pixels exist at the leaf tips.

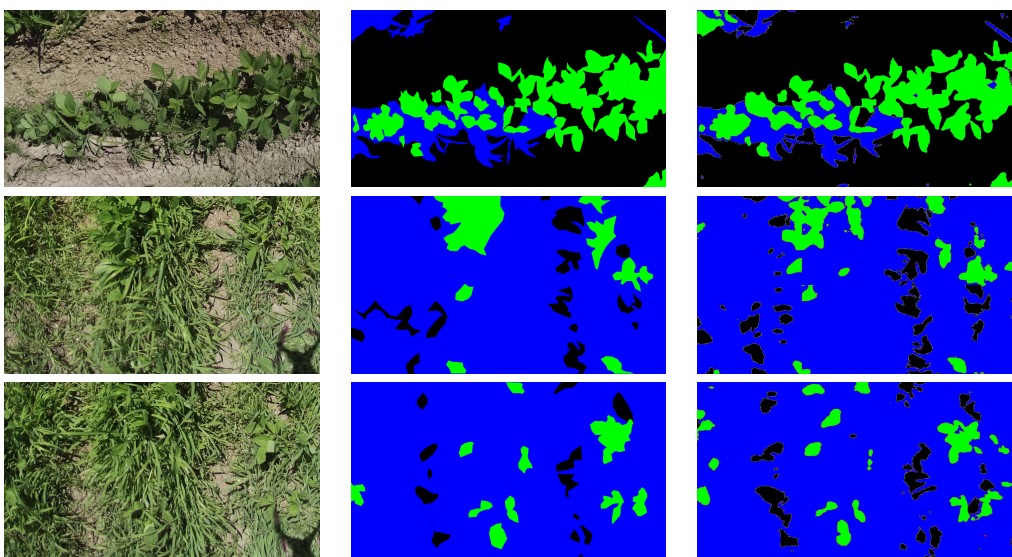

**Figure 20.** Representative successful Soybeans images: source images (**left**), GT (**center**), and results (**right**).

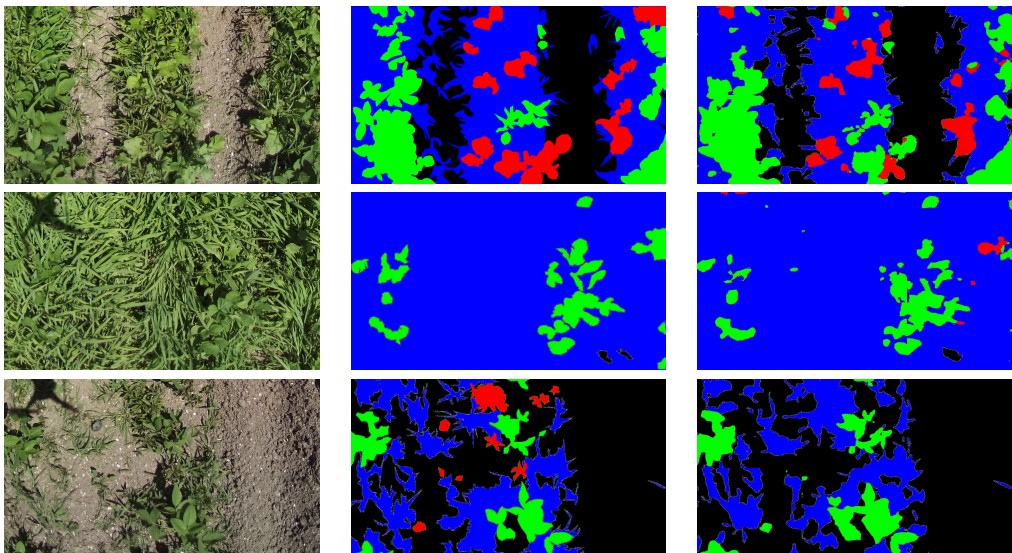

**Figure 21.** Representative failed Soybeans images: source images (**left**), GT (**center**), and results (**right**).

### 6.3. Experiment A

Experiment A provides semantic segmentation obtained using augmented images generated from the CUT as an unconditional GAN. Tables 9–11 respectively show IoU scores for Cityscapes, Rice blast, and Soybeans. The mIoU scores for Soybeans were 0.49 for the training subset and 0.20 for the test subset. The highest and lowest IoU scores were achieved, respectively, for the `road` and `traffic light` classes. The Rice blast mIoU scores were 0.76 for the training subset and 0.25 for the test subset. The highest and lowest IoU scores were achieved, respectively, for the RL and RB classes. The Soybeans mIoU scores were 0.73 for the training subset and 0.55 for the test subset. The highest and lowest IoU scores were achieved, respectively, for the PW and OW classes.

**Table 9.** IoU scores for Cityscapes in Experiment A.

| ID | 0 | 1 | 2 | 3 | 4 | 5 | 6 | 7 | 8 | 9 | 10 |
|---|---|---|---|---|---|---|---|---|---|---|---|
| Training | 0.69 | 0.65 | 0.95 | 0.83 | 0.78 | 0.50 | 0.49 | 0.10 | 0.000 | 0.14 | 0.81 |
| Test | 0.54 | 0.01 | 0.82 | 0.33 | 0.53 | 0.01 | 0.01 | 0.05 | 0.002 | 0.03 | 0.62 |
| **ID** | **11** | **12** | **13** | **14** | **15** | **16** | **17** | **18** | **19** | **20** | **Mean** |
| Training | 0.59 | 0.85 | 0.42 | 0.01 | 0.83 | 0.42 | 0.40 | 0.49 | 0.02 | 0.23 | 0.49 |
| Test | 0.08 | 0.50 | 0.06 | 0.01 | 0.53 | 0.01 | 0.02 | 0.01 | 0.01 | 0.02 | 0.20 |

**Table 10.** IoU scores for Rice blast in Experiment A.

| | BG | RL | RB | Mean |
|---|---|---|---|---|
| Training | 0.83 | 0.86 | 0.59 | 0.76 |
| Test | 0.26 | 0.49 | 0.04 | 0.25 |

**Table 11.** IoU scores for Soybeans in Experiment A.

| | BG | SL | PW | OW | Mean |
|---|---|---|---|---|---|
| Training | 0.77 | 0.81 | 0.80 | 0.55 | 0.73 |
| Test | 0.52 | 0.53 | 0.81 | 0.32 | 0.55 |

*6.4. Experiment B*

Experiment B provides semantic segmentation obtained using augmented images generated from the pix2pixHD as a conditional GAN. Tables 12–14 respectively present IoU scores for Cityscapes, Rice blast, and Soybeans. The mIoU scores for Cityscapes, Rice blast, and Soybeans were, respectively, 0.51, 0.69, and 0.76 for the training subset and 0.27, 0.42, and 0.59 for the test subset. Trends of the highest and lowest IoU scores were similar to those for Experiment A.

**Table 12.** IoU scores for Cityscapes in Experiment B.

| ID | 0 | 1 | 2 | 3 | 4 | 5 | 6 | 7 | 8 | 9 | 10 |
|---|---|---|---|---|---|---|---|---|---|---|---|
| Training | 0.69 | 0.27 | 0.94 | 0.68 | 0.81 | 0.36 | 0.34 | 0.13 | 0.03 | 0.26 | 0.83 |
| Test | 0.68 | 0.03 | 0.88 | 0.48 | 0.69 | 0.05 | 0.04 | 0.09 | 0.01 | 0.13 | 0.73 |
| **ID** | **11** | **12** | **13** | **14** | **15** | **16** | **17** | **18** | **19** | **20** | **Mean** |
| Training | 0.57 | 0.88 | 0.50 | 0.19 | 0.85 | 0.40 | 0.61 | 0.70 | 0.35 | 0.43 | 0.51 |
| Test | 0.10 | 0.63 | 0.17 | 0.03 | 0.65 | 0.03 | 0.07 | 0.01 | 0.01 | 0.15 | 0.27 |

**Table 13.** IoU scores for Rice blast in Experiment B.

| | BG | RL | RB | Mean |
|---|---|---|---|---|
| Training | 0.88 | 0.93 | 0.72 | 0.84 |
| Test | 0.52 | 0.72 | 0.15 | 0.46 |

**Table 14.** IoU scores for Soybeans in Experiment B.

| | BG | SL | PW | OW | Mean |
|---|---|---|---|---|---|
| Training | 0.78 | 0.80 | 0.80 | 0.32 | 0.68 |
| Test | 0.53 | 0.53 | 0.81 | 0.33 | 0.55 |

*6.5. Experiment C*

Experiment C provides semantic segmentation obtained using augmented images generated from CUT and pix2pixHD. Tables 15–17 respectively denote IoU scores for

Cityscapes, Rice blast, and Soybeans. The mIoU scores for Cityscapes, Rice blast, and Soybeans were, respectively, 0.47, 0.63, and 0.77 for the training subset and 0.25, 0.38, and 0.58 for the test subset. Trends of the highest and lowest IoU scores are equivalent to Experiments A and B.

**Table 15.** IoU scores for Cityscapes in Experiment C.

| ID | 0 | 1 | 2 | 3 | 4 | 5 | 6 | 7 | 8 | 9 | 10 |
|---|---|---|---|---|---|---|---|---|---|---|---|
| Training | 0.69 | 0.43 | 0.94 | 0.72 | 0.81 | 0.32 | 0.35 | 0.11 | 0.01 | 0.21 | 0.82 |
| Test | 0.66 | 0.02 | 0.88 | 0.45 | 0.68 | 0.02 | 0.04 | 0.07 | 0.01 | 0.08 | 0.71 |
| **ID** | **11** | **12** | **13** | **14** | **15** | **16** | **17** | **18** | **19** | **20** | **Mean** |
| Training | 0.57 | 0.86 | 0.45 | 0.09 | 0.83 | 0.41 | 0.36 | 0.52 | 0.14 | 0.33 | 0.47 |
| Test | 0.09 | 0.62 | 0.14 | 0.02 | 0.63 | 0.02 | 0.06 | 0.01 | 0.01 | 0.13 | 0.25 |

**Table 16.** IoU scores for Rice blast in Experiment C.

| | BG | RL | RB | Mean |
|---|---|---|---|---|
| Training | 0.62 | 0.78 | 0.47 | 0.63 |
| Test | 0.42 | 0.68 | 0.05 | 0.38 |

**Table 17.** IoU scores for Soybeans in Experiment C.

| | BG | SL | PW | OW | Mean |
|---|---|---|---|---|---|
| Training | 0.83 | 0.79 | 0.81 | 0.63 | 0.77 |
| Test | 0.52 | 0.53 | 0.80 | 0.49 | 0.58 |

*6.6. Experiment D*

Experiment D provides semantic segmentation obtained using the source images and augmented images generated from CUT and pix2pixHD. Tables 18–20 respectively present IoU scores for Cityscapes, Rice blast, and Soybeans. The mIoU scores for Cityscapes, Rice blast, and Soybeans were, respectively, 0.50, 0.69, and 0.76 for the training subset and 0.27, 0.42, and 0.59 for the test subset. Trends of the highest and lowest IoU scores are equivalent to Experiments A, B, and C.

**Table 18.** IoU scores for Cityscapes in Experiment D.

| ID | 0 | 1 | 2 | 3 | 4 | 5 | 6 | 7 | 8 | 9 | 10 |
|---|---|---|---|---|---|---|---|---|---|---|---|
| Training | 0.69 | 0.37 | 0.94 | 0.71 | 0.81 | 0.31 | 0.33 | 0.12 | 0.01 | 0.22 | 0.82 |
| Test | 0.69 | 0.04 | 0.88 | 0.47 | 0.70 | 0.03 | 0.05 | 0.08 | 0.01 | 0.10 | 0.73 |
| **ID** | **11** | **12** | **13** | **14** | **15** | **16** | **17** | **18** | **19** | **20** | **Mean** |
| Training | 0.58 | 0.87 | 0.46 | 0.20 | 0.83 | 0.51 | 0.54 | 0.58 | 0.24 | 0.36 | 0.50 |
| Test | 0.10 | 0.63 | 0.17 | 0.03 | 0.65 | 0.02 | 0.07 | 0.01 | 0.01 | 0.14 | 0.27 |

**Table 19.** IoU scores for Rice blast in Experiment D.

| | BG | RL | RB | Mean |
|---|---|---|---|---|
| Training | 0.73 | 0.82 | 0.52 | 0.69 |
| Test | 0.48 | 0.70 | 0.09 | 0.42 |

**Table 20.** IoU scores for Soybeans in Experiment D.

| | BG | SL | PW | OW | Mean |
|---|---|---|---|---|---|
| Training | 0.82 | 0.79 | 0.81 | 0.62 | 0.76 |
| Test | 0.53 | 0.52 | 0.82 | 0.48 | 0.59 |

*6.7. Overall Comparison and Discussion*

6.7.1. Cityscapes

Table 21 presents the result of a comparison of mIoU scores in each Cityscapes experiment. Experiment B produced the highest mIoU score. This result demonstrates the effectiveness of using augmented images obtained using the conditional GAN. By contrast, the mIoU score obtained from Experiment C was lower than that of the baseline. We infer that the augmented images obtained using the conditional and unconditional GANs provided no contribution to improvement of the accuracy. Experiment D results demonstrated that, using all images, including the original, improved the mIoU score.

**Table 21.** Comparison results of mIoU scores for the respective Cityscapes experiments (Bold values show maximum scores).

| Mode | Baseline | A | B | C | D |
|---|---|---|---|---|---|
| Training | 0.48 | 0.49 | **0.51** | 0.47 | 0.50 |
| Test | 0.27 | 0.20 | **0.27** | 0.25 | **0.27** |

Figure 22 depicts representative images of comparative semantic segmentation results in each Cityscapes experiment. The `road`, `car`, `vegetation`, and `building` classes were segmented correctly. False segmentation pixels occurred at the sidewalks on the left and right sides of the images. Based on these results, we infer that segmentation between `fence` and `vegetation` is a challenging task for this dataset.

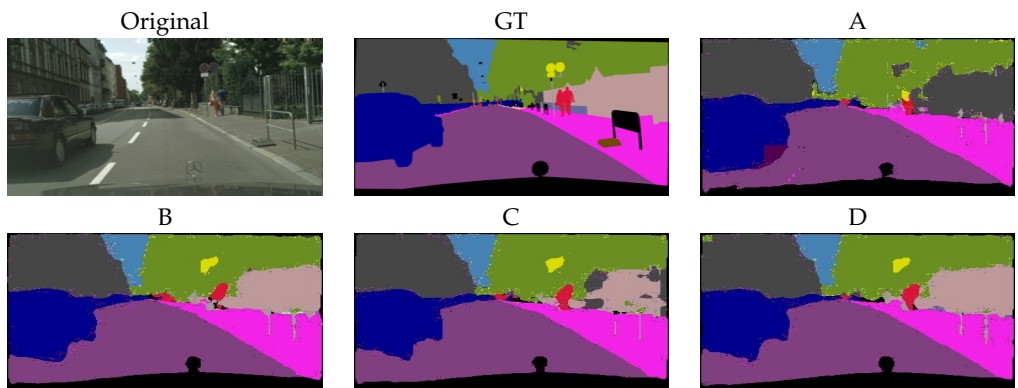

**Figure 22.** Comparison results of semantic segmentation in each Cityscapes experiment.

Figure 23 depicts results obtained from comparison of the respective experiments for the lower and upper five classes ordered by the IoU scores in Table 4. The lower five classes are `traffic light`, `pole`, `rider`, `ground`, and `traffic sign` in the training subset images and are `traffic light`, `train`, `motorcycle`, `terrain`, and `rider` in the test subset images. Two classes of `traffic light` and `rider` are common in both datasets. Results obtained from Experiment B improve accuracy in the respective classes. Although the test subset showed remarkable improvement in accuracy for the *ground* class, it dropped considerably for the `rider` class. The upper five classes are `building`, `vegetation`, `car`, `sky`, and `road`. They are `car`, `unlabeled`, `building`, `vegetation`, and `road` for the test subset. Three classes of `building`, `vegetation`, and `road` are common to both. The accuracy for the training subset exhibits no remarkable difference. For the test subset, the argument data combinations in Experiments B and D were slightly predominant. We consider that Experiments A and C contributed no accuracy improvement. However, the improved accuracy results in Experiments B and D verify a tendency that is similar to the overall results, as shown in Table 21.

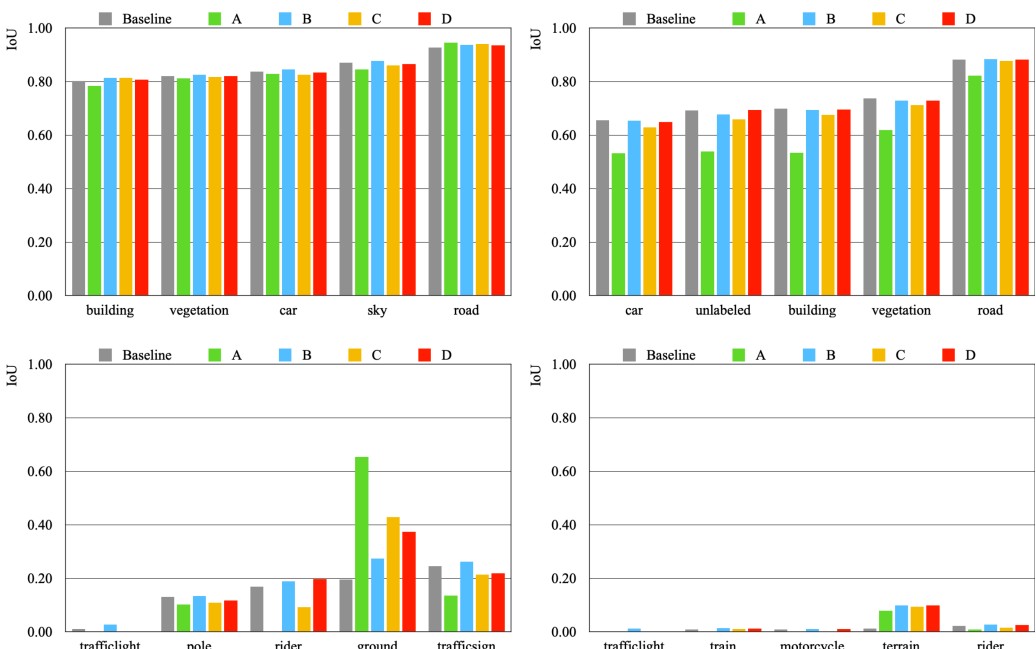

**Figure 23.** Comparison of IoU scores between the respective classes: upper five class for training subset (**upper left**), upper five class for test subset (**upper right**), lower five class for training subset (**bottom left**), and lower five class for test subset (**bottom right**).

### 6.7.2. Rice Blast

Table 22 presents a comparison of the respective results obtained for mIoU scores in experiments for Rice blast. For the training subset, mIoU scores of the four experiments were found to be lower than those of the baseline. For the test subset, mIoU scores for Experiments A, C, and D were found to be lower than the baseline score. The mIoU score for Experiment B was found to be 0.01 higher than the baseline score.

**Table 22.** Compared results of mIoU scores from the Rice blast experiments (Bold values show maximum scores).

| Mode | Baseline | A | B | C | D |
|---|---|---|---|---|---|
| Training | **0.87** | 0.76 | 0.84 | 0.63 | 0.69 |
| Test | 0.45 | 0.25 | **0.46** | 0.38 | 0.42 |

Figure 24 depicts representative images of comparative semantic segmentation results in each Rice blast experiment. Compared to other experiments, the segmentation result in Experiment B is similar to the GT images, especially in RB pixels annotated in red. By contrast, the segmentation result in Experiment A shows a trend toward incorrect segmentation of pixels in the respective classes.

Figure 25 presents compared results for the respective classes. For the training subset, no class achieved accuracy improvement compared to the baseline. For the test subset, IoU scores for RB were improved by 0.05 from 0.10 of the baseline to 0.15 in Experiment B.

### 6.7.3. Soybeans

Table 23 shows compared results of mIoU scores from the respective Soybeans experiments. For the training subset, the mIoU scores of all four experiments were found to be higher than those of the baseline. For the test subset, the mIoU scores obtained from Experiments A and B were found to be lower than those of the baseline. However, the mIoU scores in Experiments C and D were found to be 0.02 and 0.03 higher than those of the baseline for the test subset, respectively.

| Original | GT | A |
|---|---|---|

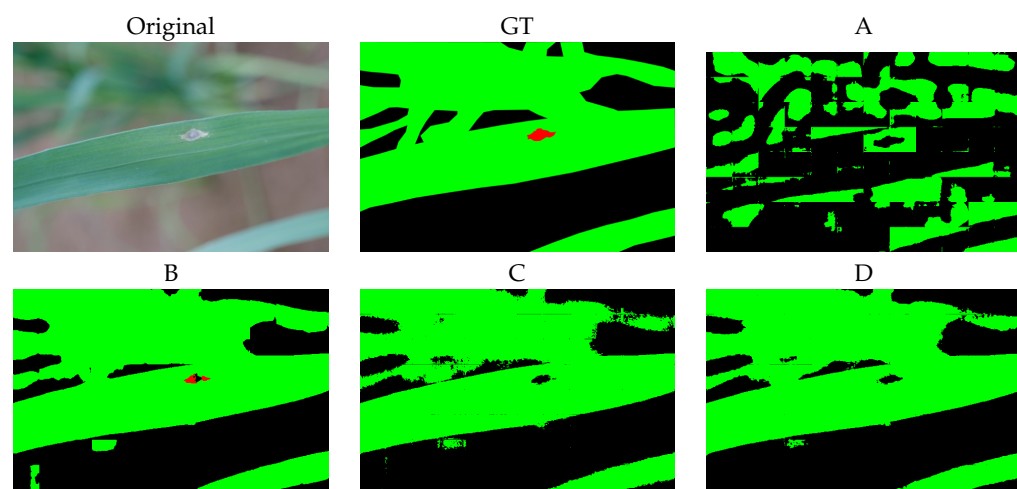

| B | C | D |
|---|---|---|

**Figure 24.** Compared results of semantic segmentation for Rice blast experiments.

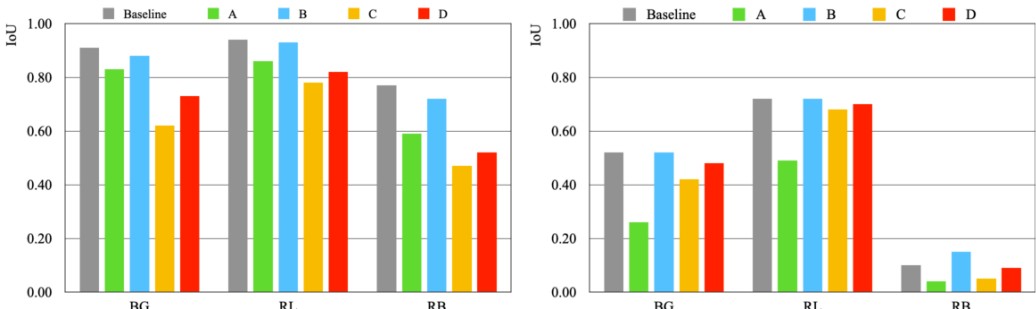

**Figure 25.** Compared results of IoU scores in the respective classes: Training dataset (**left**) and Test dataset (**right**).

**Table 23.** Compared results of mIoU scores in Soybeans experiments (Bold values show maximum scores).

| Mode | Baseline | A | B | C | D |
|---|---|---|---|---|---|
| Training | 0.59 | 0.73 | 0.68 | **0.77** | 0.76 |
| Test | 0.56 | 0.55 | 0.55 | 0.58 | **0.59** |

Figure 26 depicts representative images of comparative semantic segmentation results for each experiment conducted with Soybeans. The overall segmentation trends are similar among the four experiments. The results of the PW pixels in the bottom center panels of the images are different parts. These pixels were segmented as SL, except for the Experiment D result. However, the PW pixels were expanded in the bottom right panels of the images.

Figure 27 depicts comparison results in each class. For the training subset, all experiment results showed improved accuracy compared to the baseline. Particularly for OW, although the baseline result was 0.12, Experiments C and D results were 0.63 and 0.62. The BG and PW classes achieved comparable accuracy for the test subset. Although the IoU score for SL is lower than the mean score, the IoU scores for OW in Experiments C and D were 0.49 and 0.48, which were, respectively, 0.17 and 0.16 higher than those of the baseline score.

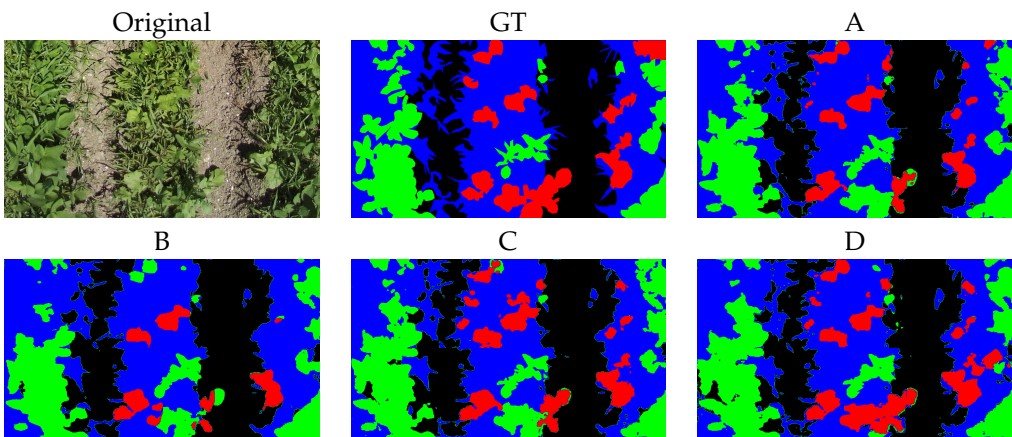

**Figure 26.** Results obtained from comparing semantic segmentation in the respective experiments with Soybeans.

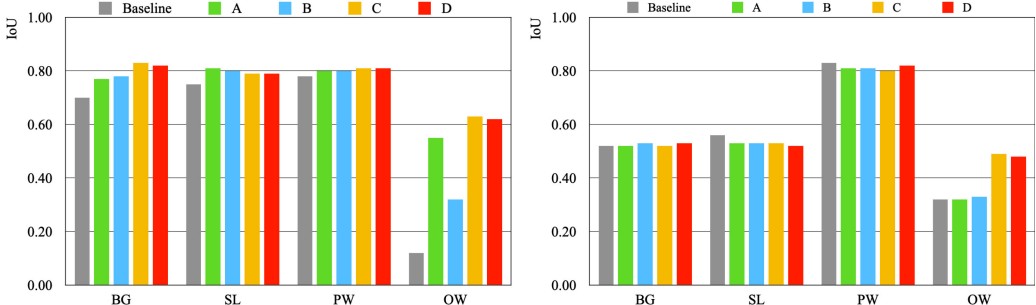

**Figure 27.** Comparison results of IoU scores in each class: training dataset (**left**) and test dataset (**right**).

## 7. Conclusions

This paper presented a data augmentation framework to improve semantic segmentation accuracy based on style transfer using GANs of two types. In particular, this study specifically addressed data generation of crop images for remote and smart farming to reduce the burden associated with image annotation. The proposed framework comprised pix2pixHD, CUT, and DeepLabV3+ as a conditional GAN, an unconditional GAN, and a semantic segmentation network. We used the Cityscapes public benchmark dataset and two originally collected benchmark datasets termed Rice blast and Soybeans. In addition, we conducted four experiments using the source images and augmented images from the combination of two GANs. The mIoU scores for the public benchmark dataset improved by 0.03 for the training subset, while remaining similar on the test subset. For the first original benchmark dataset, the mIoU scores improved by 0.01 for the test subset, while they dropped by 0.03 for the training subset. The mIoU scores for the second original benchmark dataset improved by 0.18 for the training subset and 0.03 for the test subset. These results demonstrated the effectiveness of our proposed framework using augmented images generated from two GANs in response to dataset combinations.

As future work, we must evaluate the effectiveness of our framework for use with various agricultural benchmark datasets. We will analyze the replacement of GANs and semantic segmentation backbones by considering a tradeoff between the accuracy and computational cost. We will consider the introduction of an automatic parameter optimization method for improving accuracy. Finally, we would like to collaborate in the application of our framework with a weed-removing robot or a harvesting robot to achieve remote and smart farming.

**Author Contributions:** Conceptualization, S.Y.; methodology, H.M.; software, K.T.; validation, K.T. and T.K.S.; formal analysis, K.S. (Kazuki Saruta); investigation, S.C.; resources, Y.N.; data curation, Y.N.; writing—original draft preparation, H.M.; writing—review and editing, K.S. (Kazuhito Sato); visualization, S.N.; supervision, K.S. (Kazuhito Sato); project administration, Y.N.; funding acquisition, S.Y. All authors have read and agreed to the published version of the manuscript.

**Funding:** This research was funded by Japan Society for the Promotion of Science (JSPS) KAKENHI Grant Number 21H02321.

**Institutional Review Board Statement:** Not applicable.

**Informed Consent Statement:** Not applicable.

**Data Availability Statement:** Datasets described as a result of this study are available on request to the corresponding author.

**Acknowledgments:** We would like to show our appreciation to the students at our lab for their great cooperation in the experiments.

**Conflicts of Interest:** The authors declare that they have no conflict of interest. The funders had no role in the design of the study, in the collection, analyses, or in interpretation of data, in the writing of the manuscript, or in the decision to publish the results.

## Abbreviations

The following abbreviations are used in this manuscript:

| | |
|---|---|
| BG | Background |
| BEGAN | Boundary equilibrium generative adversarial network |
| CAE | Convolutional auto encoder |
| CCL | Cycle consistency loss |
| CL | Contrastive learning |
| CMT | Conventional machine-learning |
| CUT | Contrastive unpaired translation |
| DCGAN | Deep convolutional generative adversarial network |
| DL | Deep learning |
| EBGAN | Energy-based generative adversarial network |
| ESRGAN | Enhanced super-resolution generative adversarial network |
| FID | Fréchet Inception distance |
| GAN | Generative adversarial network |
| GNSS | Global navigation satellite systems |
| GT | Ground truth |
| I2L | Image to label |
| IoT | Internet of things |
| IoU | Intersection over union |
| L2I | Label to image |
| OASIS | Only adversarial supervision for semantic image segmentation |
| OW | Other weeds |
| PGGAN | Progressive growing generative adversarial network |
| PW | Poaceae weeds |
| RB | Rice blast |
| RoI | Region of interest |
| RL | Rice leaves |
| SL | Soybean leaves |
| SPADE | Spatially adaptive denormalization |
| SSL | Self-supervised learning |
| USIS | Unsupervised paradigm for semantic image synthesis |
| ViT | Vision transformer |

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
