# Peer review of "Semantic Segmentation of Agricultural Images Based on Style Transfer Using Conditional and Unconditional Generative Adversarial Networks"

_applsci, doi:10.3390/app12157785_

Round 1

Reviewer 1 Report

This study is interesting and the whole paper is well written, some moderate revisions are shown as:

(1) The results and conclusions in the abstract of this paper are not clear enough and seem to be too bried.

(2) The second paragraph of "related studies" is too long. Maybe divided into 2.1   2,2   2,3,etc, is ok.

(3) The text content should be above the figure in the section 3.1.

(4) Do not have too many acronyms, and you must pay attention to the readability of your writing.

(5) There are too many tables, some tables with few contents can be merged or deleted or described by text directly.

Author Response

Thank you very much for reviewing our manuscript.
We appreciate your helpful comments and suggestions.
The attached file contains our responses and a revised manuscript.

Reviewer 2 Report

The current research study presents state-of-the-art data augmentation methods based on image style transfer and GANs. The authors analyze present framework for semantic segmentation combined with data augmentation using two GAN models of different mechanisms and their setup properties including benchmark datasets of three types and evaluation metrics. 

The study presents a topic of relevance for the readers of the journal, but however it requires a revision before publishing.

Please refer to the following points, comments and suggestions that are given in the attached file

Author Response

(The authors gave the same response as above.)
